# The Development of a Rabies Virus-Vectored Vaccine against *Borrelia burgdorferi*, Targeting BBI39

**DOI:** 10.3390/vaccines12010078

**Published:** 2024-01-12

**Authors:** Shantel Rios, Bibek Bhattachan, Kruthi Vavilikolanu, Chrysoula Kitsou, Utpal Pal, Matthias J. Schnell

**Affiliations:** 1Department of Microbiology and Immunology, Sidney Kimmel Medical College, Thomas Jefferson University, Philadelphia, PA 19107, USA; 2Jefferson Vaccine Center, Sidney Kimmel Medical College, Thomas Jefferson University, Philadelphia, PA 19107, USA; 3Department of Veterinary Medicine, University of Maryland, College Park, MD 20740, USA; bbek2020@umd.edu (B.B.); kvavilik@umd.edu (K.V.); ckitsou@umd.edu (C.K.)

**Keywords:** Lyme disease vaccine, *Borrelia burgdorferi*, rabies virus, viral vaccine vector

## Abstract

Lyme disease (LD) is the most common tick-borne illness in the United States (U.S.), Europe, and Asia. *Borrelia burgdorferi*, a spirochete bacterium transmitted by the tick vector *Ixodes scapularis*, causes LD in the U.S. If untreated, Lyme arthritis, heart block, and meningitis can occur. Given the absence of a human Lyme disease vaccine, we developed a vaccine using the rabies virus (RABV) vaccine vector BNSP333 and an outer surface borrelial protein, BBI39. BBI39 was previously utilized as a recombinant protein vaccine and was protective in challenge experiments; therefore, we decided to utilize this protective antigen in a rabies virus-vectored vaccine against *Borrelia burgdorferi.* To incorporate BBI39 into the RABV virion, we generated a chimeric BBI39 antigen, BBI39_RVG_, by fusing BBI39 with the final amino acids of the RABV glycoprotein by molecular cloning and viral recovery with reverse transcription genetics. Here, we have demonstrated that the BBI39_RVG_ antigen was incorporated into the RABV virion via immunofluorescence and Western blot analysis. Mice vaccinated with our BPL inactivated RABV-BBI39_RVG_ (BNSP333-BBI39_RVG_) vaccine induced high amounts of BBI39-specific antibodies, which were maintained long-term, up to eight months post-vaccination. The BBI39 antibodies neutralized *Borrelia* in vaccinated mice when challenged with *Borrelia burgdorferi* by either syringe injection or infected ticks and they reduced the Lyme disease pathology of arthritis in infected mouse joints. Overall, the RABV-based LD vaccine induced more and longer-term antibodies compared to the recombinant protein vaccine. This resulted in lower borrelial RNA in RABV-based vaccinated mice compared to recombinant protein vaccinated mice. The results of this study indicate the successful use of BBI39 as a vaccine antigen and RABV as a vaccine vector for LD.

## 1. Background

Lyme disease (LD), also called Lyme borreliosis, is the most common vector-borne illness in the United States (U.S.), Europe, and Asia, with approximately 476,000 estimated new cases diagnosed annually in the U.S. alone [1]. The etiologic agent of LD in North America is the spirochete pathogen *Borrelia burgdorferi* (Bb) and the tick vector *Ixodes scapularis*, or the deer tick. Bb begins its infection at the tick bite site in the host’s skin. An erythema migrans rash, a clinical skin lesion, can appear on the skin due to the spread of Bb from the initial bite site [2]. Symptoms of early stage LD mimic a viral-like illness and include fever, headache, fatigue, muscle and joint aches, and swollen lymph nodes [3]. However, 20–30% of patients do not display the rash, which can vary in morphological features, and patients can test negative for Bb with a rash present, which challenges clinicians when making an LD diagnosis [2,4]. Frequent misdiagnoses further exacerbate the disease into late-stage LD where Bb travels to distant organs, including the joints, heart, and central nervous system [4,5]. This can cause complications including arthritis with joint pain and swelling, heart block, and inflammation of the brain and spinal cord [4]. Currently, the only treatment for LD is antibiotics; however, after treatment and Bb clearance, patients can still develop Post-treatment Lyme Disease Syndrome (PTLDS), a chronic disease in patients previously diagnosed with LD [6]. These complications indicate the need for a human LD vaccine.

Despite efforts to create a vaccine [7,8], no effective FDA-approved human LD vaccine exists. The rise in global temperatures has caused the tick population to double in the past two decades [9]. The CDC predicts that a continued rise in the tick population will increase LD cases [9]. Additionally, the economic burden of LD is significant; the current annual cost of treatment is estimated to be USD 1.3 million per year for the U.S. healthcare system [3]. A rise in cases will continue to increase costs, whereas a preventative vaccine would provide a long-term solution for preventing LD [9].

Previously, the FDA approved an LD vaccine, LYMErix, a recombinant OspA protein vaccine adjuvanted with alum [10]. OspA has been shown to protect against Bb and other strains of *Borrelia*. LYMErix decreased LD rates by 68% within its first year and 92% efficacy in its second year on the market [11]. A couple years after the vaccine was put on the market, there was a reported linkage of arthritis development in OspA-vaccinated patients with treatment-resistant Lyme arthritis. In addition, vaccine sales declined and the vaccine was taken off the market [11]. OspA remains a target antigen in LD vaccines such as in Vanguard crLyme [12], a dog vaccine, and VLA15, a human vaccine developed by Pfizer, currently in phase III clinical trials (NCT05477524) [7]. However, there is currently no LD vaccine on the market.

To avoid these possible adverse events from OspA immunization, we developed a non-OspA LD vaccine with the protective outer surface protein BBI39 [13]. BBI39 is an outer surface protein produced on Bb with unknown functions [13]. BBI39 is in the paralogous gene family (pgf), found on plasmid lp54 in Bb. Pgfs are differentially expressed and regulated by temperature, pH, and various other intrinsic factors of the tick or mammalian hosts. BBI39 is expressed on the surface of Bb within the tick and in early host infection in the skin. This compares to OspA, which is solely produced while Bb resides in the tick and is downregulated when entering the host [14]. A previous study found that mice vaccinated with the recombinant protein BBI39 induced BBI39-specific antibodies, depleted the borrelial load, and reduced LD pathogenesis in the heart and joints [13]. Therefore, we used BBI39 as our target antigen for our rabies virus (RABV)-vectored LD vaccine. 

To create an effective LD vaccine, a highly immunogenic vaccine vector is crucial. RABV vaccine vectors produce a long-lasting humoral immune response [15]. These vaccine vectors are safe because they are highly attenuated and can be made into inactivated vaccines [16,17,18,19]. This study utilized the recombinant RABV vector BNSP333, derived from SAD-B19, an attenuated wildlife rabies vaccine strain [20]. BNSP333 has been further attenuated by an arginine-to-glutamine mutation at amino acid position 333 of the RABV glycoprotein (G). This mutation further reduces the neurovirulence of the RABV vector and increases its safety profile [21,22]. BNSP333 has a simple genome with only five proteins, making it easy to manipulate and allowing for the addition of stably expressed foreign antigens [23]. The RABV vaccine is highly immunogenic; it induces long-term immunity and a type-1-biased immune response, making it an ideal vector for protection against Bb [24,25]. RABV has been used as a vaccine vector against various infectious diseases, including SARS-CoV-2 (CoraVax) [19], Lassa virus (LassaRab) [26], Ebola virus (FiloRab1) [27], and Crimean Congo Hemorrhagic Fever Virus [28]. Some of these vaccines have been tested in nonhuman primates (NHPs) and human studies, further highlighting the safety and efficacy of this vaccine vector [29].

In this study, we utilized the attenuated RABV vaccine vector with the borrelial outer surface protein BBI39. We believed that the addition of the RABV vaccine vector would induce high antibody titers that last longer-term than a recombinant protein vaccine. BBI39 is an ideal vaccine antigen because it has been shown to protect against Bb infection as a recombinant protein immunization. Therefore, when BBI39 is vectored with RABV, there will be better immunogenicity and protection against Bb. We demonstrate that the RABV vector successfully incorporated the chimeric BBI39_RVG_ antigen into the RABV virion, which is necessary for antibody production. Mice immunized with our candidate vaccine, BNSP333-BBI39_RVG_, induced high and long-term anti-BBI39 antibody titers with a Th-1-biased immune response compared to the recombinant protein vaccine. In addition, we studied the adjuvant effects of PHAD-SE to determine whether this adjuvant was ideal for an LD vaccine. Finally, we show that vaccinated mice inhibited Bb infection during both syringe inoculum and Bb-infected tick challenge experiments. We show that the RABV-vectored BBI39_RVG_ vaccine is an ideal vaccine candidate against LD.

## 2. Materials and Methods

### 2.1. Borrelia burgdorferi, Cell Lines, Mice, and Ticks

*Borrelia burgdorferi* strain B31, grown in Barbour-Stoenner-Kelly-H (BSK-H) medium complete (supplemented with 6% rabbit serum) (Sigma-Aldrich B8291, St. Louis, MO, USA), was used in this study. Bacteria were grown at 33 °C without CO_2_ in 5 mL bacterial culture tubes. *Ixodes scapularis* ticks were maintained in a colony within the Pal lab at the University of Maryland. Ticks were subjected to a microinjection of Bb to perform challenge experiments with infected ticks. BSR and Vero (CCL81) cells were obtained from ATCC and cultured in 1X DMEM (Corning Cat# 10-013-CV, Corning, NY, USA) supplemented with 5% fetal bovine serum (FBS) and 1% penicillin-streptomycin. Cells were maintained at 37 °C with 5% CO_2_. Six- to eight-week-old C3H/HeN mice were purchased from Charles River Laboratories. All animal experiments were performed under the guidelines of the Institutional Animal Care and Use Committee and Institutional Bio-safety Committee of Thomas Jefferson University. 

### 2.2. cDNA Molecular Cloning of Vaccine Vectors

We inserted BBI39 and BBI39_RVG_, both synthesized by Genscript (Piscataway, NJ, USA) in pUC57 vectors, into BsiWI and NheI restriction sites of the BNSP333 RABV vaccine vector [18,30] via T4 Ligation (New England Biolabs catalog #: M0202, Ipswitch, MA, USA). This included BNSP333-BBI39 and BNSP333-BBI39_RVG_. BBI39_RVG_ included the final 51 amino acids of the ectodomain (ED51), transmembrane domain (TM), and cytoplasmic domain (CD) of RABV-G, all in the gene synthesis. JM109 *E. coli* cells were used during molecular cloning under ampicillin resistance. Once plasmids were synthesized, they were sent out for sequencing to Azenta (Walham, MA, USA). We utilized forward (5′-GGAGGTCGACTAAAGAGATCTCACATAC-3′) and reverse (5′-TTCTTCAGCCATCTCAAGATCGGCCAGAC) primers to sequence BBI39 between RABV-N and RABV-P. We also used forward (5′-GTTATGGTGCCATTAAACCGCTG-3′) and reverse (5′-TCTCCAGGATCGATCGAGCATCTT-3′) primers to sequence RABV-G to determine whether the 333 mutation was still viable in the glycoprotein before virus recovery.

### 2.3. Recovery of Recombinant Rabies Viruses

Recombinant RABV vectors were recovered on BSR cells in the above-listed conditions. X-tremeGENE 9 transfection reagent (Millipore Sigma XTTG9-RO, Burlington, MA, USA) in Opti-MEM was utilized to transfect BSR cells in 6-well plates with full-length BNSP333 cDNA along with T7 RNA polymerase, RABV nucleoprotein, phosphoprotein, glycoprotein, and polymerase cDNA plasmids. Supernatants were harvested on day four post-transfection and overlayed on seeded BSR cells in 12-well plates. After 48 h, cells were subjected to fixing by acetone and staining with a GFP stain against RABV-N (FujiRebio, Cat# 800-092, Malvern, PA, USA).

### 2.4. Viral Growth, Titration, Purification, and Inactivation

Once the viruses were recovered, they were grown on Vero CCL81 cells in viral production serum-free medium (VP-SFM) (Thermo Fisher Scientific, Waltham, MA, USA) supplemented with 5% Glutamax, 1% penicillin-streptomycin, and 1% HEPES buffer. Cells were infected at an MOI of 0.01 in either a T175 flask or 2-stack chambers (Corning, Corning, NY, USA). Supernatant collections occurred 5 days post-infection and every 3 days afterward, for a total of 17 days. For titration, RABVs were overlayed on VeroCCL81 cells with a starting dilution of 1:10 and diluted 10-fold in a 96-well plate in triplicate. After 48 h, cells were fixed with acetone and stained with RABV-N GFP stain to determine the foci-forming units (ffu)/mL using fluorescence microscopy. 

Viruses were filtered through 0.45 µm PES membrane filters (Nalgene, Conway, NH, USA) and concentrated down to 50 mL for purification. Viruses were purified over 20% sucrose cushion and ultracentrifuged at 25,000 rpm in a SW32 rotor for 1.5 h. Viral particles were resuspended in TEN buffer + 2% sucrose and inactivated with 50 µL per mg of particles with β-propiolactone (BPL, Millipore Sigma, Cat# P5648, Burlington, VT, USA) in cold culture grade water. The level of inactivation was verified by inoculating Vero CCL81 cells over three passages with 10 µg of BPL-inactivated virions. 

### 2.5. Immunofluorescence

Vero CCL81 cells were seeded on coverslips in 1x DMEM supplemented with 5% FBS and 1% penicillin-streptomycin at 5 × 10^5^ cells per well. On the same day, cells were infected with BNSP333-BBI39_RVG_, BNSP333-BBI39, and BNSP333 at an MOI of 0.05. Cells were kept at 34 °C for 72 h and then stained. Cells were fixed in 2% paraformaldehyde (PFA) in 1X PBS for 30 min for surface-stained cells and 15 min for intracellular-stained cells. Then, 2% PFA and 0.01% TritonX were added to intracellular stained cells for another 15 min. After fixing, cells were blocked in PBS with 5% FBS for 1 h. Following washing with PBS three times, cells were stained with 1:200 primary antibody for 1 h at room temperature (RT) in PBS with 1% FBS. Primary antibodies included polyclonal mouse anti-BBI39 IgG and human anti-RABV-G 4C12 (provided by Scott Dessain, Lankenau Institute for Medical Research, Wynnewood, PA, USA). Cells were washed with PBS and incubated with secondary antibody in PBS with 1% FBS for 45 min at RT. Secondary antibodies utilized for fluorescent staining were anti-mouse Cy3 and anti-human Cy2. Stained cells were then washed 5 times with PBS and mounted on slides with mounting media containing DAPI (ProLong™ Glass Antifade Mountant, Invitrogen, cat#: P36980, Waltham, MA, USA). Slides were stored for 24–48 h in the dark at RT to dry. Slides were visualized with a Nikon A1R confocal microscope. Images were analyzed by Fuji. Red (Cy3) BBI39 staining was changed to magenta by FIJI ImageJ, Version 1.

### 2.6. Cell Lysate Preparation for Western Blot

For infected cell lysates, 1 × 10^6^ BSR cells were infected at an MOI of 5 for 72 h at 34 °C in a 6-well plate. Cells were washed twice with cold PBS, and 1 mL of RIPA lysis buffer + 1X protease inhibitor (ThermoFisher Halt™ Protease Inhibitor Cocktail 100X, cat#: 78430, Norristown, PA, USA) was added to lyse infected cells. After 5 min of lysis, cells were centrifuged at 14,000 rpm for 1 min. The supernatant was then subjected to a BCA assay to determine the concentration of the proteins from the cell lysates. Finally, the concentration of cell lysates was adjusted to 10 µg/µL in urea sample buffer containing 2-mercaptoethanol.

### 2.7. Western Blot

Infected cell lysates, recombinant proteins, and purified viral particles were subjected to Western blot analysis. Lysates and particles were denatured in urea sample buffer and reduced with 2-mercaptoethanol. Samples were boiled for 10 min at 95 °C. In total, 30 µg of cell lysates, 20 ng of BBI39 recombinant protein, and 1 µg of sucrose-purified virions were separated on the gel. Gels were run at 150 V for about 1.5 h in 1X Laemmli buffer. Proteins were transferred to nitrocellulose membranes for 1 h at 90 V in 1X Towbin transfer buffer. Electrophoresis of gels and transfer of proteins were achieved using BioRad Western blot equipment. After transfer, the membranes were blocked with 5% milk in 1X PBST for 1 h at RT. The primary antibodies used for probing included polyclonal mouse anti-BBI39 IgG and human anti-RABV-G 4C12 (provided by Scott Dessain, Lankenau Institute for Medical Research, Wynnewood, PA, USA). Blots were probed overnight with 5% BSA in PBS at 4 °C. The following day, blots were probed with secondary antibody, which included horseradish peroxidase (HRP)-conjugated anti-mouse (Jackson ImmunoResearch, 115-035-146, West Grove, PA, USA) at 1:5000 or human IgG (SouthernBiotech, 2040-05, Birmingham, AL, USA) at 1:20,000, diluted in 1X PBST for 1 h at RT. Proteins were detected with SuperSignal West Dura Chemiluminescent substrate (Thermofisher cat#: A38554, Norristown, PA, USA) and imaged on a FlourChem R system (ProteinSimple, San Jose, CA, USA).

### 2.8. Production of Recombinant Proteins for Western Blots, Immunizations, and ELISA

BBI39 was purified using a plasmid from the Utpal Pal laboratory in University Park, Maryland. BBI39 plasmid transformed into JM109 *E. coli* cells and grew into a 1 L bacterial culture in LB broth with ampicillin. Bacteria were induced with 0.3 mM of ITPG overnight at RT. The following day, bacteria were centrifuged at 4000 rpm for 30 min at 4 °C. The pellet was resuspended in 20 mL of PBS with 1% TritonX. Then, for further lysis, 0.1 mg/mL of Lysozyme was added to the resuspension. After a 30 min incubation on ice, bacterial cells were further lysed by sonication for 7 min. Lysed bacterial cells were centrifuged at 4000 rpm for 30 min at 4 °C. BBI39 was purified using Glutathione Agarose (Pierce™, Thermofisher cat#: 16100, Norristown, PA, USA). The GST tag was removed by 80 units of precision protease (GenScript cat#: Z02799, Piscataway, NJ, USA) overnight at 4 °C in PCB buffer (50 mM Tris pH 7.0, 150 mM NaCl, 1 mM EDTA, 1 mM DTT, mixed in water). Purification of the protein was further characterized by Western blot, as listed above.

RABV glycoprotein (G) was produced by stripping the glycoprotein from rVSV-ΔG-RABV-G-GFP virions. BEAS-2B cells were infected with rVSV-ΔG-RABV-G-GFP in OptiPRO SFM at an MOI of 0.01. Once all cells were lysed, supernatants were concentrated and ultracentrifuged through a 20% sucrose cushion at 25,000 rpm for 1.5 h at 4 °C. Viral pellets were then resuspended in β-Octyl-glucopyranoside (OGP) and stripped by ultracentrifugation at 45,000 rpm for 1.5 h in an SW55Ti rotor at 4 °C. Supernatants were collected, frozen in small aliquots, and characterized by Western blot and ELISA. 

### 2.9. Immunizations

Groups of five 6–8-week-old female and male C3H/HeN mice purchased from Charles River Laboratories were immunized intramuscularly (I.M.) with 10 µg of inactivated RABV virions or 1 × 10^4^ ffu/mL of live attenuated virus. Five females were used for initial immunogenicity experiments. Five females and five males were used for challenge experiments. Inactivated vaccines were formulated either unadjuvanted or with 5 µg of synthetic monophosphoryl Lipid-A (MPLA), 3D(6 A)-PHAD, in a squalene oil-in-water emulsion (PHAD-SE) adjuvant. Each immunization contained 100 µL, with 50 µL injected into each hind leg of each mouse. All mice were primed on day 0 and boosted on day 28. Serum for further testing was collected via retroorbital bleeds while mice were under isoflurane anesthesia on days 0, 14, 28, and 56 for short-term experiments with the addition of days 112, 168, and 224 for long-term experiments.

### 2.10. Anti-BBI39 and Anti-RABV-G ELISA

Total and isotype subclass IgG antibody responses were determined by indirect ELISA. We purified recombinant RABV-G and BBI39, described above, and utilized these proteins to coat Immulon 4 HBX 96-well flat-bottom microtiter plates. Plates were coated with antigens overnight at 4 °C in 15 mM Na_2_CO_3_, 35 mM NaHCO_3_ coating buffer. BBI39 antigen was utilized at 500 ng/well and RABV-G at 50 ng/well. Post-incubation, plates were washed three times with PBS containing 0.05% Tween20 (PBST) and blocked in 5% milk for 2 h at RT. Plates were washed and primary antibody dilution buffer, containing 0.5% BSA and 0.05% NaN_3_ in PBST, was added at 100 µL per plate. Mouse sera, at a 1:50 starting dilution, was further diluted 3-fold down the plates. Plates were incubated overnight at 4 °C with primary antibody. The following day, plates were washed, and 100 µL of secondary antibody diluted in PBST was added. The secondary antibodies used in this study were horseradish peroxidase-conjugated goat anti-mouse IgG-Fc (Jackson ImmunoResearch, Cat# 115-005-008 West Grove, PA, USA); IgG2a (Jackson ImmunoResearch Cat# 115-035-206 West Grove, PA, USA); or IgG1 (Jackson ImmunoResearch Cat# 115-035-205 West Grove, PA, USA). All secondary antibodies were diluted to a concentration of 80 ng/mL for BBI39 ELISAs and 25 ng/mL for RABV-G ELISAs. After incubation for 2 h at RT, plates were washed and then developed with 200 µL/well of o-Phenylenediamine Dihydrochloride substrate (ThermoFisher, Norristown, PA, USA) for 15 min. The reaction was stopped with 3M H_2_SO_4_. The optical density (OD) was determined at 490 nm (experimental) and 630 nm (background) on a BioTek ELx800 plate reader (BioTek, Winooski, VT, USA) using Gen5 software to determine the delta values between the experimental and background readings. ELISA data were analyzed in GraphPad Prism 9 software to determine the EC50 values of antibodies in the mouse sera. 

### 2.11. Borreliacidal Assay

*Borrelia burgdorferi*, strain B31, was seeded at 1 × 10^5^ spirochetes/mL in 96-well round-bottom plates with a 1:10 dilution of heat-inactivated mouse sera or 100 ug/mL of LA-2 antibody (Absolute Antibody, Ab01070-3.0, Boston, MA, USA) with 1:10 guinea pig complement sera (Sigma S1639) diluted in BSK-H complete media (Sigma-Aldrich B8291, St. Louis, MO, USA) for a total of 100 μL. Mouse sera were heat-inactivated at 56 °C for 30 min and tested in duplicate. Borrelia was incubated at 33 °C with mouse sera for 48 h in a 96-well round-bottom plate. In total, 1 µL of each well was added to 1 mL of BSK media in Eppendorf tubes. After 7 days, each replicate was counted using a Nikon dark-field microscope with a Petroff-Hausser counting chamber (Hausser Scientific, Cat#: 3900, Horsham, PA, USA). Borrelial counts were analyzed in GraphPad Prism 9 software to determine if mouse sera inhibited bacterial growth by borreliacidal activity.

### 2.12. RABV Neutralization Determined by Rapid Fluorescent-Focus Inhibition Tests (RFFITs)

RFFITs were performed to identify RABV-neutralizing antibodies as described previously [19]. Mouse sera were heat inactivated at 56 °C for 30 min. BSR cells were seeded at 25,000 cells/well and cultured in DMEM with 5% FBS and 1% penicillin-streptomycin in 96-well flat-bottom plates. Individual mouse sera, collected at day 56, were diluted 3-fold with a starting dilution of 1:50. The WHO standard of rabies IgG was used at a starting dilution of 2 international units (IU)/mL. After the dilution of sera, CVS11, a challenge virus strain of RABV, was added to each well at a titer to achieve 90% infection of BSR cells. The virus and antibody mixture was incubated in a 96-well round-bottom plate for 1 h at 34 °C. After incubation, 105 µL of sera/virus mixture was added to BSR cells and incubated at 34 °C for 24 h. Cells were fixed with 80% acetone and stained with FAD stain against RABV nucleoprotein. Stained cells were assessed for their percentage of viral infection by fluorescent microscopy. The Reed–Muench method was utilized to calculate 50% endpoint titers. Titers were converted to IU/mL via comparison to the WHO standard.

### 2.13. Borrelia burgdorferi Challenge via Needle Injection

Immunized and unimmunized mice were subjected to challenge with *Borrelia burgdorferi*, strain B31, via the needle route. Mice were injected with 1 × 10^5^ spirochetes/100 µL intradermally using an insulin needle. Bacteria were grown in BSK-H complete medium (Sigma-Aldrich B8291, St. Louis, MO, USA) at 33 °C before being counted using Nikon dark-field microscopy with a Petroff-Hausser counting chamber (Hausser Scientific, Cat#: 3900). Bacteria were diluted in BSK media in Eppendorf tubes and kept at RT before injection. 

At 21 days post-infection, skin (ear), tibiotarsi joints, heart, and bladders were harvested from infected mice aseptically and these were subjected to RNA extraction and qPCR (described below) or further analysis under dark-field microscopy. Organs were placed in 1 mL BSK-H medium and incubated at 33 °C. Every 2 weeks, for 8 weeks, organs were analyzed using dark-field microscopy for the qualitative analysis of *Borrelia* in each organ’s supernatant. 

Blood was also collected from challenged mice and subjected to Western blot analysis to determine whether each mouse was successfully challenged with Borrelia. *Borrelia burgdorferi* was grown in 50 mL to 1 × 10^8^ spirochetes/mL. Bacteria were centrifuged at 4000× *g* for 20 min at 4 °C. Borrelial pellets were washed five times with 1 mL of PBS. After each wash, the lysates were centrifuged in Eppendorf tubes at 14,000× *g* for 1 min. Borrelial lysates were subjected to BCA assay to determine the final concentration. Lysates were diluted to a final concentration of 1.5 µg/10 µL in 1X urea sample buffer. Aliquots were frozen at −80 °C or used for Western blot analysis. Lysates were denatured for 10 min at 95 °C, and gels were run and transferred as described above. For primary antibody, individual mouse sera were diluted to 1:1000 in 5% BSA and added to strips of nitrocellulose membrane with Borrelial lysates transferred on each strip. Primary antibody was incubated overnight and further processed as listed above. Individual strips were imaged at the same time to test whether challenged or unchallenged mouse sera responded to borrelial lysates on blots. 

### 2.14. Borrelia burgdorferi Challenge via Ixodes scapularis

Groups of six mice were immunized as described earlier. On day 56 post primary immunization, mice were challenged with infected *Ixodes scapularis* nymphs (5 ticks/mouse). After three weeks of infection, mice were euthanized, and the Bb burden within mouse organs was assessed by qRT-PCR (described below). Skin, heart, and joints were cultured in BSK-H media as described above. 

### 2.15. RNA Extraction and Quantitative Real-Time Polymerase Chain Reaction (RT-qPCR)

Organs from challenged mice were harvested and placed in 1 mL TRIzol Reagent in 2 mL RNase/DNase free Omni tubes (Omni International, Kennesaw, GA, USA), which contained beads for homogenization. Bladders were collected in Omni tubes with 1.4 mm ceramic beads (Omni International, Cat#: 19-627D, Kennesaw, GA, USA). Hearts and skin (ear) were collected in Omni tubes with 2.8 mm ceramic beads (Omni International, Cat#: 19-628D, Kennesaw, GA, USA). Joints were collected in Omni tubes with 2.8 mm metal beads (Omni International, Cat#: 19-620D, Kennesaw, GA, USA). Tubes were homogenized using a Omni Bead Ruptor for 90 s. Homogenates were frozen at −80 °C for RNA extraction. RNA was extracted from whole organs using the TRIzol Reagent phase separation protocol. RNA extraction was performed using the PureLink RNA Mini Kit (Invitrogen, Waltham, MA, USA). The quantity and quality of RNA extracted was measured using NanoDrop (Thermofisher, Norristown, PA, USA).

Borrelia was quantified by RT-qPCR. FlaB and mouse B-actin primer probes were designed for use with TaqMan Fast Virus 1 Step Master Mix reagent (ThermoFisher Norristown, PA, USA), using 5 µL of extracted RNA from each mouse organ. Primer probes were ordered from ThermoFisher (Norristown, PA, USA). FlaB was amplified by forward primer (5′-TTGCTGATCAAGCTCAATATAACCA-3′) and reverse primer (5′-GCATCGCTTTCAGGGTCTCAA-3′) with a probe quenched with FAM fluorescent dye (5′-AGAACAGCTGAAGAGCTTGGAATGCAGCCTGCAAAAATTAACACA-3′). Mouse B-actin was amplified by forward primer (5′-AGAGGGAAATCGTGCGTGAC-3′) and reverse primer (5′-ACGGCCAGGTCATCACTATTG-3′) with a probe quenched with VIC fluorescent dye (5′-CAAAGAGAAGCTGTGCTATGTTGCTCTAGACTTCGAGCAGGAGAT-3′). The reaction was set up for a fast-cycling mode with the following cycling protocol: 1 cycle for 5 min at 50  °C, 1 cycle for 20 s at 95  °C, and 40 cycles of 95  °C for 3 s and 60  °C for 30 s. The reactions were run on a Step One Plus qPCR machine.

### 2.16. Histological Analysis

Joints from challenged mice were collected from each group three weeks after infection. Joints were fixed in 4% paraformaldehyde (PFA) and stained with hematoxylin–eosin (H&E) stain. Signs of arthritis were evaluated, as described elsewhere, in a blinded manner [31]. 

### 2.17. Statistical Analysis

For ELISA, log-transformed 50% effective concentration (EC50) values were determined by plotting against the delta OD value (OD [490 nm]-OD [630 nm]) on GraphPad Prism 9 software. For all statistical analyses, a one-way ANOVA with post hoc Tukey HSD test was performed on log-transformed data.

## 3. Results

### 3.1. BBI39 Vaccine Designs

In this study, we utilized the well-established attenuated RABV vector BNSP333 [19,25,26,27,28,29] (Figure 1a). The gene encoding the borrelial antigen, BBI39, was inserted in BNSP333 at the second position between the RABV nucleoprotein (N) and phosphoprotein (P) and named BNSP333-BBI39 (Figure 1b). This vaccine determined the immunogenicity of BBI39 after immunization with the live viral vector BNSP333-BBI39. We also developed the BNSP333-BBI39_RVG_ vaccine to incorporate a chimeric BBI39 protein into the RABV virion. This vaccine can be produced as a killed vaccine due to the incorporation of the antigen into the RABV virion. The chimeric BBI39 antigen contained the RABV-G Igκ signal sequence (SS) fused to the 3′ end of the BBI39 nucleotide sequence, followed by the sequences for the last 51 amino acids of the ectodomain (ED51), the transmembrane domain (TM), and cytoplasmic domain (CD) of RABV-G (Figure 1c). These sections are called the RVG tail. We previously showed that these additions promote the translocation of the foreign protein to the endoplasmic reticulum (ER) and transport it to the surface of an infected cell, allowing for BBI39 incorporation into the budding RABV virion [19,28].

### 3.2. Characterization of Newly Developed BBI39 Vaccines

Infectious recombinant viruses were recovered via previously established methods [32,33]. To confirm BBI39 expression, we characterized these viruses by immunofluorescence staining and Western blotting analysis. First, we conducted the intracellular immunostaining of infected cells for the BBI39 protein (magenta) and RABV-G (green) with their respective antibodies (Figure 2a). BBI39 was expressed intracellularly by both the BNSP333-BBI39_RVG_ and BNSP333-BBI39 viruses. In addition, the proteins of infected cell lysates were separated by SDS-PAGE and probed for both BBI39 and RABV-G using a Western blot. We confirmed the expression of BBI39 by both viruses, BNSP333-BBI39_RVG_ (50 kDa) and BNSP333-BBI39 (30 kDa) (Figure 2b). This includes RABV-G at about 65 kDa (Figure 2b).

To verify the transport of BBI39 to the surface of an infected cell via the RVG tail, we performed another immunofluorescence assay with surface-stained cells. BNSP333-BBI39_RVG_ successfully translocated BBI39 to the surface of the cell, but BNSP333-BBI39 did not (Figure 2c).

After the successful translocation of BBI39 to the surface of the cell, the protein was incorporated into the budding BNSP333-BBI39_RVG_ viral virion. Sucrose-purified virions were separated on an SDS-PAGE protein gel and probed for using Western blotting. The SDS-PAGE protein gel displayed all RABV proteins (L, G, N, P, and M) within the RABV virion (Appendix A). Western immunoblotting detected the incorporation of BBI39 and RABV-G in BNSP333-BBI39_RVG_ virions; however, BNSP333-BBI39 did not incorporate BBI39 (Figure 2d). While both viruses produce BBI39 intracellularly (Figure 2a,b), only the chimeric BBI39_RVG_ protein, expressed by BNSP333-BBI39_RVG_, was detected on the surface of the infected cells and incorporated into the RABV virion (Figure 2c,d).

### 3.3. Immunogenicity of RABV-BBI39 Vaccines

After the characterization of the recombinant viruses, we studied the immunogenicity of the vaccines in C3H/HeN mice, a well-characterized model for studying LD in mice [13]. When these mice are infected with *Borrelia*, they show the symptoms of LD observed in humans, such as Lyme arthritis and carditis [13].

Five C3H/HeN mice per group were immunized with 10^4^ foci-forming units (ffu) of live BNSP333-BBI39 or BNSP333-BBI39_RVG_. Since BNSP33-BBI39_RVG_ incorporates BBI39 into the virion, we also tested it as an inactivated vaccine, both with and without the adjuvant PHAD-SE [24]. Several studies have demonstrated that PHAD-SE induces a Th1-biased immune response [24,25]. This should induce borreliacidal antibodies that can bind, complement, and protect against *Borrelia burgdorferi*, as shown in other studies [34,35,36]. Mice were immunized on day 0 and boosted on day 28 (Figure 3a). As a positive control, we included the previously tested recombinant BBI39 protein, which depleted Bb in both a tick and syringe challenge [13]. Since BBI39 protein immunization was previously adjuvanted with Freund’s adjuvant, an adjuvant not sanctioned for human use, we included the adjuvant PHAD-SE. Finally, as an RABV vector control, we used FiloRab1 [24], a well-studied BNSP333-vectored vaccine incorporating the Ebola virus glycoprotein. Serum was collected on days 0, 14, 28, and 56 post initial immunization and analyzed for the presence of IgG antibodies against BBI39 and RABV-G by ELISA (Figure 3b,c).

The detection of anti-BBI39 IgG responses show the induction of antibody responses as early as 14 days after the first immunization in both the inactivated vaccine groups (BNSP33-BBI39_RVG_, with and without adjuvant) and the recombinant protein BBI39 group, with a 1-fold lower titer seen in the live BNSP333-BBI39_RVG_ group (Figure 3b). BBI39-specific antibody responses were not observed in the live BNSP333-BBI39 vaccine group (Figure 3b). These responses demonstrated a slight 1-fold increase in the adjuvanted groups on day 28 compared to day 14. Serum titers increased almost 100-fold on day 56 post-boost for all groups vaccinated with BNSP333-BBI39_RVG_ and recombinant protein immunized groups. Overall, on day 56, BNSP333-BBI39_RVG_ + PHAD-SE vaccinated mice had the highest BBI39 and RABV-G IgG titers compared to all other groups (Figure 3b,c). Live BNSP333-BBI39 vaccinated mice continued to not elicit anti-BBI39 antibodies after the boost on day 56; however, anti-RABV-G antibodies confirmed a successful vaccination with BNSP333-BBI39 (Figure 3c). This demonstrates that cell surface expression and the incorporation of BBI39 into the RABV virion is critical to induce an antibody response against the borrelial antigen. Since BNSP333-BBI39 did not elicit antibodies against BBI39, this vaccine was excluded from further studies.

Overall, inactivated BNSP333-BBI39_RVG_ + PHAD-SE induced a 1.5-fold higher anti-BBI39 IgG response than recombinant protein BBI39 + PHAD-SE immunized mice (Figure 3d). In addition, to analyze sex as a scientific variable, we included 5 female and 5 male mice in our study (*n* = 10) (Figure 3d). We detected a 1.5-fold difference between females and males in the recombinant protein immunized groups and less than a 1-fold difference in the RABV-vectored vaccinated groups (Appendix A), but these differences were not statistically significant. However, the stronger humoral immune responses induced by the RABV-based vectors compared to the recombinant protein vaccine (Figure 3b–d) indicated the potential benefits of the RABV vector platform compared to the recombinant protein vaccine platform.

Finally, we analyzed the type of antibody responses via the BBI39 IgG isotype responses (Figure 3e). We detected IgG2a, a type-1-associated immune response antibody, and IgG1, a type-2-associated antibody by ELISA. As previously demonstrated [24,25], BNSP333-BBI39_RVG_ + PHAD-SE induced a 2-fold higher IgG2a response than the unadjuvanted and recombinant protein immunized groups (Figure 3e). The protein-vaccinated mice demonstrated a balanced response between IgG2a and IgG1, whereas RABV-vectored vaccines, both with and without adjuvant, induced a more significant skew towards a type-1-associated immune response, previously shown to protect against *Borrelia* [34,35] (Figure 3e).

### 3.4. Neutralization of Borrelia burgdorferi and Rabies Virus by BBI39 Vaccines

Next, we determined whether these antibodies were functional against both *Borrelia burgdorferi* and RABV. We conducted a borreliacidal assay to determine the neutralization of Bb and found that mouse sera against BBI39, with the addition of guinea pig complement, delayed the growth of *Borrelia* in culture by nearly 10-fold. A significantly greater growth delay was found in the RABV-vectored groups, especially with the addition of PHAD-SE (Figure 4a). This inhibition is compared to the positive control, LA-2 antibody, which is a protective monoclonal antibody [37]. However, the sera from recombinant protein-immunized mice, even with adjuvant, did not significantly deplete Bb in vitro compared to FiloRab1-vaccinated mice, naïve mice and in no mouse sera (NMS) (Figure 4a). These data indicate the potential benefits of the RABV vector platform compared to recombinant protein vaccines.

We also analyzed the potential of the RABV-vectored vaccines to neutralize infectious RABV in an RFFIT. Protection against both *Borrelia* and RABV would be another advantage of this vaccine. Serum from all RABV virus-vectored vaccinated mice contained a neutralizing titer of above 0.5 IU/mL, the WHO-accepted protective antibody level against RABV (Figure 4b). BNSP333-BBI39_RVG_ + PHAD-SE adjuvant induced 3-fold higher neutralizing titers compared to unadjuvanted groups. These levels were compared to the control group, FiloRab1 + PHAD-SE, which was previously shown to produce high neutralization effects against RABV [24]. Therefore, the RABV-based BBI39 vaccine can induce both borreliacidal and RABV-neutralizing antibodies in vitro.

### 3.5. Reduction in Borrelia burgdorferi in Vaccinated Mice from Syringe Inoculation

To determine the ability of this vaccine to prevent LD and borrelial infection, we conducted a syringe inoculum challenge experiment with *Borrelia burgdorferi* strain B31. Vaccinated and unvaccinated C3H mice, female and male (*n* = 10), were injected intradermally with 1 × 10^5^ spirochetes/mL by needle injection on day 56 (Figure 5a). Successful Bb infection was confirmed by Western blot of *Borrelia* lysates and probing with individual mouse sera (Appendix A). At 21 days post-infection, joints, hearts, skin, and bladders were collected from infected mice to determine the borrelial RNA load by qRT-PCR. For analysis, the FlaB CT values were normalized to mouse Beta-actin CT values. Mice vaccinated with BNSP333-BBI39_RVG_, with and without PHAD-SE, and BBI39 recombinant protein showed a significant reduction in their borrelial load in all organs when compared to FiloRab1 mice (Figure 5b–e). BNSP333-BBI39_RVG_ + PHAD-SE showed a significant decrease in Bb in the skin compared to unadjuvanted BNSP333-BBI39_RVG_ and recombinant protein-vaccinated mice (Figure 5b). The skin was the only organ with significant differences between vaccinated groups. Since BBI39 is only produced while Bb is in the tick and in early host infection, we expected to see significant differences in vaccinated groups. However, BNSP333-BBI39_RVG_ + PHAD-SE trended lower in Bb loads in the joint and heart than the other vaccinated groups; however, this did not reach significance (Figure 5c,d). All vaccinated mice had successfully depleted Bb in their hearts, joints, skin, and bladder when compared to FiloRab1-vaccinated mice, however, they were still infected.

We also observed reduced *Borrelia* in infected mouse organs via a culture in BSK medium. In total, 15/15 FiloRab1-vaccinated mice had Bb in their hearts, joints, and skin. However, we observed a depletion in the vaccinated mice, as displayed in Appendix A. The BNSP333-BBI39_RVG_, adjuvanted and unadjuvanted, vaccinated mice consistently had less Bb in their mouse organs compared to the recombinant protein-immunized mice. This parallels the qRT-PCR data (Figure 5).

### 3.6. The BNSP333-BBI39_RVG_ Vaccine Protects against B. burgdorferi Transmission and Disease in an Infected Tick Challenge

We demonstrated that BNSP333-BBI39_RVG_-vaccinated mice can successfully reduce their Bb burden from syringe inoculum. To assess the protective efficacy, we completed a challenge using the natural route of spirochete transmission, via *I. scapularis* ticks. BNSP333-BBI39_RVG_ + PHAD-SE-vaccinated and FiloRab1 control mice (*n* = 6/vaccine group) were subjected to the infected tick challenge, using Bb-infected ticks (*n* = 5 nymphs/mouse), on day 56 post vaccination (Figure 6a). The skin and joints from infected mice were collected 21 days post challenge for qRT-PCR of the borrelial burden. The data in Figure 6b,c show that BNSP333-BBI39_RVG_ immunization reduced the Bb burden in mouse skin (5-fold) and joints (3-fold) compared to the control group, FiloRab1, during the infected tick challenge. The RABV-based vaccine reduced the Bb load more than the recombinant protein vaccine, by almost 2-fold more in the skin and 1-fold in the joints. In addition, the mouse joints from the tick challenge were analyzed for arthritis via histopathology. BNSP333-BBI39_RVG_-vaccinated mouse joints showed significantly lower arthritis scores than FiloRab1-vaccinated mice (Figure 6d,e). Recombinant protein-immunized mice did not significantly lower their arthritis scores. Therefore, we observed that BNSP333-BBI39_RVG_ + PHAD-SE can successfully reduce LD pathology in an infected tick challenge.

### 3.7. RABV-BBI39 Vaccines Are Immunogenic and Induce Long-Term Protection

Vaccines must provide long-term efficacy; therefore, we conducted a long-term vaccination and syringe challenge experiment with BBI39-vaccinated mice. Antibody EC50 titers were determined by ELISA until 8 months post initial vaccination (Figure 7a). We noted about a 1-fold depletion of anti-BBI39 antibodies after day 56; however, they remained steady from 4–8 months post immunization. We observed the lowest waning antibody titers in the BNSP333-BBI39_RVG_ + PHAD-SE group and the highest in the recombinant protein-vaccinated group after 8 months (Figure 7b). RABV-vectored groups maintained a high skew towards a type-1 immune response, with high IgG2a for vaccines with and without adjuvant. This is compared to the recombinant protein with a more balanced immune response, seen previously at 2 months (Figure 3e). Of note, we saw little to no IgG1 in unadjuvanted BNSP333-BBI39_RVG_-vaccinated mice (Figure 7c).

Long-term vaccinated mice were subjected to a Bb challenge via syringe inoculation. As described earlier, 21 days post challenge, the infected organs were collected and analyzed by qRT-PCR (Figure 7d–g). We observed lower borrelial loads (skin, joint, and heart) from all BBI39-vaccinated mice with a consistently lower Bb load in BNSP333-BBI39_RVG_, adjuvanted and unadjuvanted, vaccinated mice compared to recombinant protein-immunized mice (Figure 7d–f). Adjuvanted and unadjuvanted mice displayed a similar reduction in Bb; however, in unadjuvanted mice, Bb loads trended lower in all organs (Figure 7d–g).

Finally, the presence of Bb from infected mouse organs was observed via dark-field microscopy in BSK culture (Appendix A). We observed a reduction in *Borrelia* in infected mouse organs in all mice vaccinated with BBI39 vaccines compared to the FiloRab1 control, as displayed in Appendix A.

Our results show that antibody titers and type-1-biased immunity are higher in the RABV-immunized mice, formulated with and without adjuvant, when compared to recombinant protein immunization. The increase in type-1-biased immunity further led to a lower Bb burden in BNSP333-BBI39_RVG_-vaccinated mice compared to both recombinant protein and unvaccinated (FiloRab1) mice.

## 4. Discussion

In this study, we utilized BBI39 in the RABV vaccine vector BNSP333 and found that BBI39 can be incorporated into the RABV virion with the addition of an RVG tail. The incorporation of BBI39 in BNSP333-BBI39_RVG_ induces anti-BBI39 IgG antibodies in vaccinated mice; however, the unincorporated vaccine, BNSP333-BBI39, does not. All RABV-vectored vaccines produced anti-RABV-G antibodies. BNSP333-BBI39_RVG_, especially with the adjuvant PHAD-SE, can induce a type-1-associated immune response via the change in antibody isotypes. This leads to *Borrelia* and RABV neutralization in vitro. However, the recombinant protein vaccine, although adjuvanted with PHAD-SE, induces a more balanced type1/2 immune response, which was seen in this study and in previous studies [13,38]. BNSP333-BBI39_RVG_-vaccinated mice successfully depleted Bb in syringe and tick challenges more than the recombinant protein-immunized mice and eliminated LD pathogenesis.

In previous studies, OspA was the most utilized vaccine antigen [7,12,34,38,39]. While OspA is protective, it was found to contain a similar epitope to the human leukocyte function-associated antigen-1 (hLFA-1α_L322–340_) (OspA_165–183_) [40]. It was discovered that patients with treatment-resistant arthritis contain the MHC class II HLA DR4B1*0401 or DR4B1*0101 gene. Therefore, their T cells cross-react to the human leukocyte function-associated antigen-1 (hLFA-1α_L322–340_) and OspA_165–183_, producing an inflammatory response in the joints. In addition, other factors including low efficacy, the need for boosters for sufficient neutralizing antibody titers, and a lack of studies in children were reasons why the vaccine was taken off the market [11]. Low efficacy and the need for boosters could correlate with the recombinant protein vaccine platform used in LymeRix, which was also seen with the recombinant protein immunization from the long-term experiments in this study and a previous study [41]. Ultimately, the company removed the vaccine from the market in 2002 [42]. To prevent these complications from arising, we utilized BBI39, a different surface protein on *Borrelia* previously seen to be protective against LD [13].

Previous platforms utilized to create an LD vaccine include recombinant protein(s) [7,12,13,34,39], viral vectors [43], DNA [44], and mRNA vaccines [38,45]. However, recombinant proteins have been highly utilized for LD vaccine platforms, including LYMErix [39], VLA15 [7], and Vanguard crLyme [12]. Recombinant proteins have low and short-lived immunogenicity, requiring adjuvant and multiple inoculations with yearly revaccination [41]. Viral vaccine vectors, including RABV BNSP333, have been widely studied as efficacious, long-term options [25,26,29], but have rarely been studied as an LD vaccine. One group attempted to use Newcastle disease virus (NDV) as a vaccine vector for OspC [43], another protective borrelial antigen. However, this vaccine vector amounted to low antibody titers in mice and an insignificant depletion of Bb in various organs. Another concern with utilizing a viral vector for an LD vaccine is that the Bb proteins are glycosylated differently by *Borrelia* than when produced by mammalian cells, which is how a viral vector is developed. This could change the immune response to a bacterial protein induced by a viral vector [46]. In this study, we showed that the borrelial antigen BBI39 must be incorporated into the RABV virion to elicit high-titer anti-BBI39 antibodies. When the borrelial antigen does not include the RVG tail, BBI39 is not incorporated into the RABV virion. Without this incorporation, the protein stays inside the cell due to RABV’s non-cytolytic nature, and the antigen is not presented to the immune response to elicit antibodies. With the addition of the RVG tail to BBI39_RVG_, the antigen is incorporated into the RABV virion, and high-titer antibodies are elicited. This protects against Bb and the inhibition of the pathogenesis of LD. We observed the glycosylation of BBI39; however, this antigen is still protective in the viral vector. In fact, we saw even greater protection from the viral vector of vaccinated mice compared to the recombinant protein.

In addition, the long-term efficacy of BNSP333 with BBI39 and other vaccine antigens [25] demonstrates an ideal platform against LD. A previous study on the BBI39 recombinant protein did not include long-term immunization experiments [13]; however, our study showed a greater waning immunity of the recombinant protein up to 8 months post vaccination. In addition to waning immunity from recombinant proteins, mRNA have shown waning immunity to various antigens, requiring multiple boosts [47]. Long-term vaccination and challenge studies were not completed in the mRNA LD vaccine study [38,48]. However, we showed that with one boost of the BNSP333-BBI39_RVG_ vaccine, long-term antibody responses were maintained 8 months post vaccination. In addition, a previous study showed that BNSP333 can maintain antibody titers for up to one year and produce antibody-secreting cells in the spleen and bone marrow after only one boost, both with and without adjuvant [24]. Viral vectors, including RABV, are great candidates for developing an LD vaccine.

Previous studies demonstrated that antibodies are the main correlate of protection to prevent LD [49,50]. Type-1-associated antibodies such as IgG2a are ideal for Bb neutralization [34,36]. We saw a greater IgG2a induction and Bb neutralization in vitro from BNSP333-BBI39_RVG_ + PHAD-SE-vaccinated mice than the recombinant protein immunization with adjuvant. Therefore, RABV with adjuvant provided greater protection against Bb in vaccinated mice compared to recombinant protein-vaccinated mice. These data are compared to a previous study utilizing a PHAD-liposome particle bound with OspA that demonstrated high antibody titers, a skew towards Th1 immunity, and borreliacidal effects, which resulted in the depletion of Bb in ticks [34]. Although a challenge study was not conducted, these results correlate to the immunogenicity and borreliacidal activities in our study. This study also found long-term effects from their vaccine, aligning with our study. However, alum is widely used in the formulation of LD vaccines, including LYMErix [39] and VLA15 [7], and is known to induce a type-2-associated immune response [24], which is not ideal for Bb protection [51]. The induction of a type-1-associated immune response from the viral vector and adjuvant PHAD-SE is an ideal formulation for developing a successful LD vaccine.

In this study, we utilized a syringe challenge and infected tick challenge. This is unlike other studies that only utilized syringe inoculation [38]. The immune evasion strategies of the tick’s salivary proteins, among other strategies utilized by Bb to enter the host, are not present in a syringe challenge. Although the syringe challenge is the most feasible when a tick colony is not available, this should be considered when developing other LD vaccines. In our syringe challenge, we identified significant protection against BNSP333-BBI39_RVG_. However, in the tick challenge, Bb was depleted much less. Therefore, the tick challenge is ideal when analyzing the efficacy of a potential LD vaccine.

To further study BNSP333-BBI39_RVG_ as a vaccine candidate, other mouse models of vaccination should be studied, such as C57BL/6 and BALB/C mice, which are less inflammatory mouse models compared to C3H/HeN. This could further evaluate the differences between adjuvanted and unadjuvanted groups. In addition, non-human primates should be utilized for protective efficacy with this vaccine. Further preclinical testing to determine the inhibition of disease pathogenesis, such as arthritis and carditis, should be completed. This includes long-term challenge experiments or keeping vaccinated and challenged mice for longer periods than 21 days post challenge. This could show whether the lower borrelial burdens in vaccinated mice could continue to prevent LD pathogenesis. Since LD is highly inflammatory, the immune profile of vaccinated and unvaccinated mice, such as cytokine profiles after vaccination and challenge, should also be studied. A greater understanding of their impact on antibody titers and passive transfer experiments will elucidate the necessity of specific antibody titers during Bb infection to prevent LD. Testing this vaccine against other strains of *Borrelia*, such as *Borrelia afzelii* and *Borrelia garinii*, which are strains seen in LD patients from Europe and Asia, may determine whether the BBI39 vaccination is reliable for these strains. Finally, since we did not see sterile immunity with the BNSP333-BBI39_RVG_ vaccine in either challenge experiment, future research could study different borrelial antigens in combination with the BBI39 vaccine. Targeting the bacteria with different surface proteins could induce higher immunity against Bb. Studies should incorporate the foreign antigen into the RABV virion and use adjuvant PHAD-SE to create another RABV-based Bb vaccine. Other antigens studied for LD vaccines, such as OspA without the hLFA epitope, OspC, and other previously studied Bb or tick antigens, could be tested in another vaccine.

## 5. Conclusions

In this study, we created a vaccine, BNSP333-BBI39-RVG, which targets BBI39, a surface protein on Bb. This study is the second to utilize BBI39 as a vaccine antigen and the first to utilize RABV as a vaccine vector for LD. We demonstrated that, to produce antibodies against BBI39, the bacterial antigen must be incorporated into the RABV virion with the RVG tail to create the chimeric protein BBI39_RVG_. Both anti-BBI39 and anti-RABV-G IgG antibodies are elicited against in this vaccine, which thereby contains neutralizing functions against both Bb and RABV in vitro. Thus, vaccinated mice can successfully deplete the Bb load during syringe and tick challenges. Although these mice are still infected, the LD pathology is depleted. In addition, RABV-vectored vaccines can induce greater immunity and protect against Bb long-term, up to 8 months post immunization, in mice. Long-term protection is necessary when creating a successful vaccine.

We showed the advantages of a viral-vectored vaccine compared to recombinant protein vaccines. Viral-vectored vaccines induce greater type-1 immunity while recombinant protein vaccines produce a more balanced type-1/2 immunity. A type-1-biased immunity from the viral vector can kill Bb more efficiently than a recombinant protein vaccine in mice. In addition, PHAD-SE is an ideal adjuvant in the formulation of an LD vaccine. We saw a higher depletion of Bb in PHAD-SE-vaccinated mice compared to unvaccinated mice. In mice vaccinated with adjuvant, we observed higher antibody production overall. The results from this study will help further research to develop an effective human LD vaccine.

## Figures and Tables

**Figure 1 vaccines-12-00078-f001:**
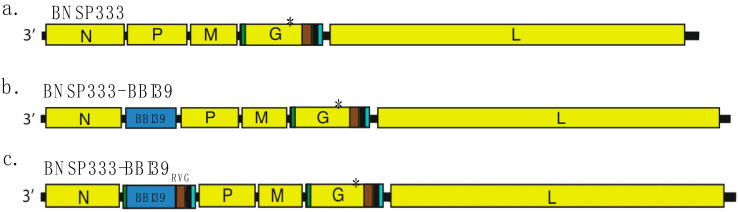
RABV-vectored vaccine constructs containing BBI39. (**a**) Rabies virus (RABV)-vectored vaccine constructs containing BBI39. BNSP333 is the parental RABV vector, with rabies nucleoprotein (N), phosphoprotein (P), matrix protein (M), glycoprotein (G), and RNA-dependent RNA polymerase (L). Highlighted in the glycoprotein is the signal peptide (Ig-kappa sequence) (green), the final 51 amino acids of the ectodomain (brown), the transmembrane domain (black), and the cytoplasmic domain (blue). (*) Indicates a point mutation at amino acid position 333 from arginine to glutamic acid. (**b**,**c**) RABV-based constructs containing BBI39. (**b**) Non-anchored construct with BBI39 placed between RABV-N and RABV-P, creating BNSP333-BBI39. (**c**) BBI39_RVG_ with the RABV-G additions to the gene, creating BNSP333-BBI39_RVG_.

**Figure 2 vaccines-12-00078-f002:**
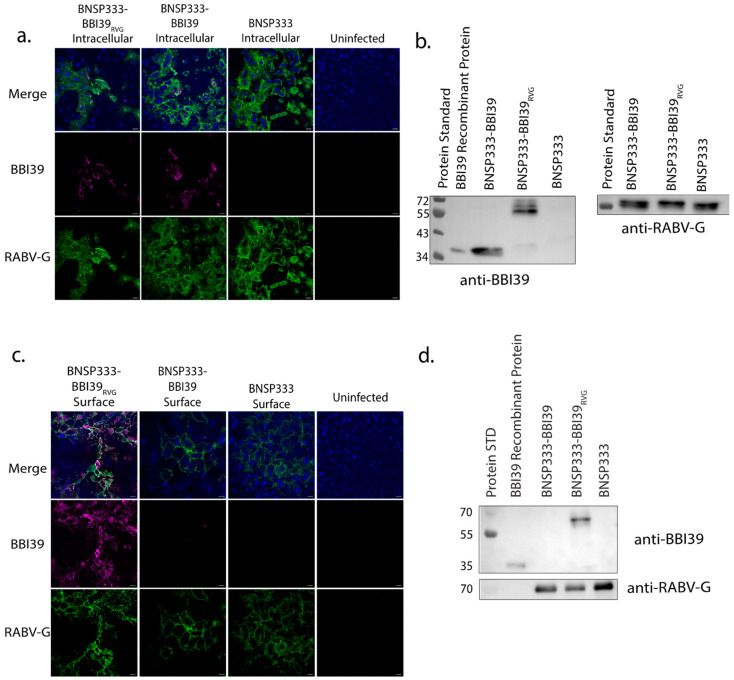
RABV can incorporate the chimeric borrelial protein BBI39_RVG_. Characterization of RABV-vectored vaccine viruses via immunofluorescence (**a**,**c**) and Western blot (**b**,**d**). For immunofluorescence, VEROCCL81 cells were infected at an MOI of 0.05 for 72 h and fixed and permeabilized for intracellular (**a**) and fixed only for surface staining (**c**). Cells were stained with human anti-RABV-G (4C12) (green) and polyclonal mouse anti-BBI39 (magenta) and mounted with mounting media containing DAPI (blue). Images were taken at 40X magnification, and scale bars represent 10 μm. Western blot of cell lysates (**b**) of VERO-CCL81 cells infected at an MOI of 0.05 for 72 h that were probed for BBI39 and RABV-G. Western blots of sucrose-purified virions (**d**) that were probed for BBI39 and RABV-G. Blots were probed with either polyclonal mouse-anti-BBI39 (produced by the Pal lab at University of Maryland) or 4C12 human anti-RABV-G monoclonal antibody (provided by Scott Dessain, Lankenau Institute for Medical Research, Wynnewood, PA) (**b**,**d**).

**Figure 3 vaccines-12-00078-f003:**
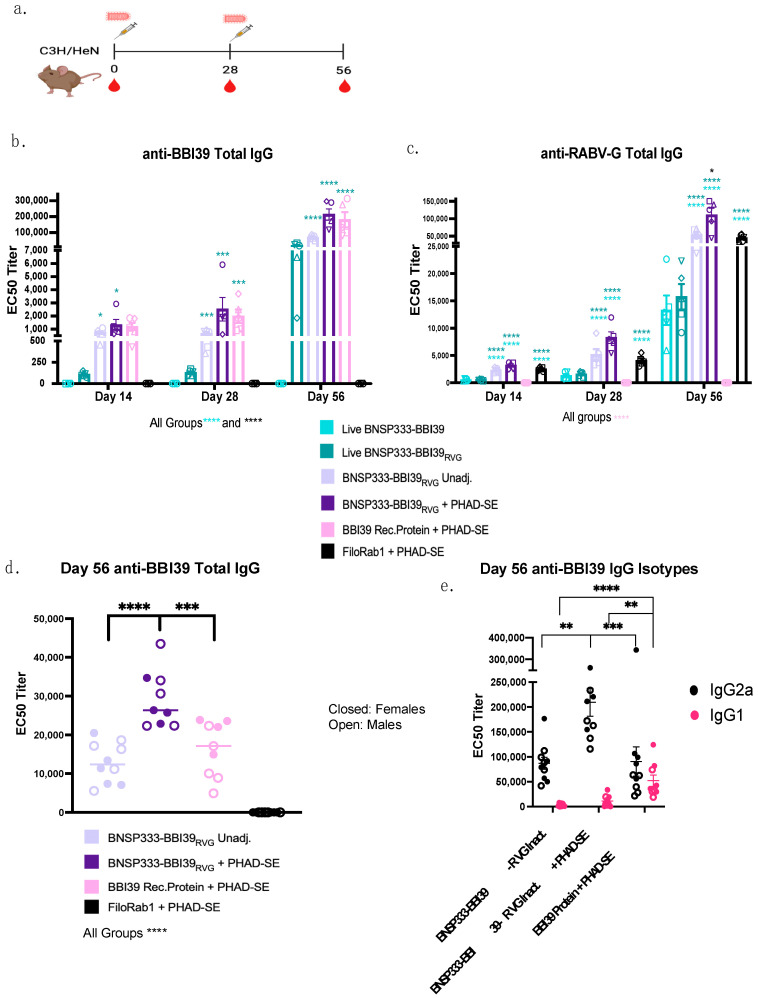
BNSP333-BBI39_RVG_ elicits anti-BBI39 and anti-RABV-G antibodies with type-1-biased immunity. ELISA anti-BBI39 and anti-RABV-G antibody half maximal effective concentration (EC50) titers from vaccinated mouse sera. (**a**) Vaccination schedule of C3H/HeN mice. Mice were primed on day 0 and boosted on day 28 with RABV-based vaccines. Blood drops show when to collect sera for testing. Anti-BBI39 (**b**) and anti-RABV-G (**c**) total IgG EC50 titers from day 14–56 determined from individual mouse serum ELISA curves (*n* = 5 females). Various shapes represent different mice in that vaccinated group. (**d**) Day 56 EC50 titers with inactivated groups only, demonstrating female and male total IgG titers (*n* = 10/vaccine group, 5 females and 5 males). (**e**) Day 56 anti-BBI39 IgG isotype ELISAs. Statistics are represented by the color of the bars of each vaccinated group. Error bars represent the SEM. Statistics were calculated by one-way ANOVA with post hoc Tukey’s test of log-transformed data. *p* > 0.05 non-significant (ns), *p* < 0.0332 (*), *p* < 0.0021 (**), *p* < 0.0002 (***), *p* < 0.0001 (****).

**Figure 4 vaccines-12-00078-f004:**
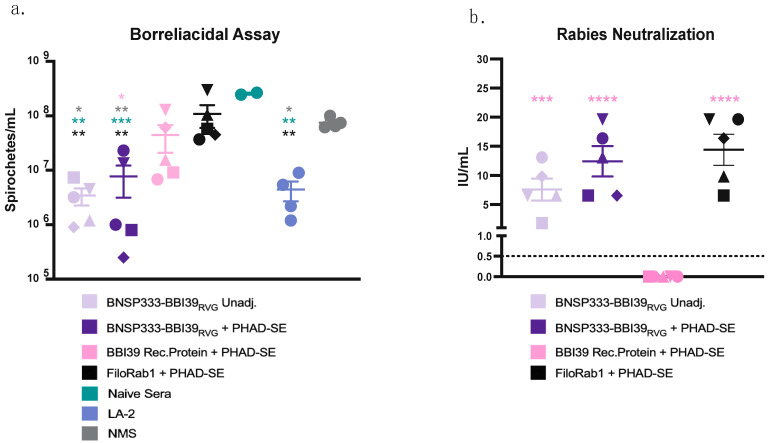
BNSP333-BBI39_RVG_ can successfully neutralize *Borrelia burgdorferi* and RABV in vitro. (**a**) Borreliacidal assay conducted with sera from vaccinated mice (5 female mice per group), naïve sera, LA-2 protective monoclonal antibody, and no mouse sera (NMS). Bb titers counted 7 days after the addition of mouse sera and guinea pig complement sera to culture. Various shapes represent different mice in that vaccinated group. (**b**) Rabies neutralization assay (RFFIT) containing sera from vaccinated mice. IU/mL titers were calculated by comparison to the WHO standard. Statistics are represented by the color of the bars of each vaccinated group. Error bars represent the SEM. Statistics were calculated by one-way ANOVA with post hoc Tukey’s test of log-transformed data. *p* > 0.05 non-significant (ns), *p* < 0.0332 (*), *p* < 0.0021 (**), *p* < 0.0002 (***), *p* < 0.0001 (****).

**Figure 5 vaccines-12-00078-f005:**
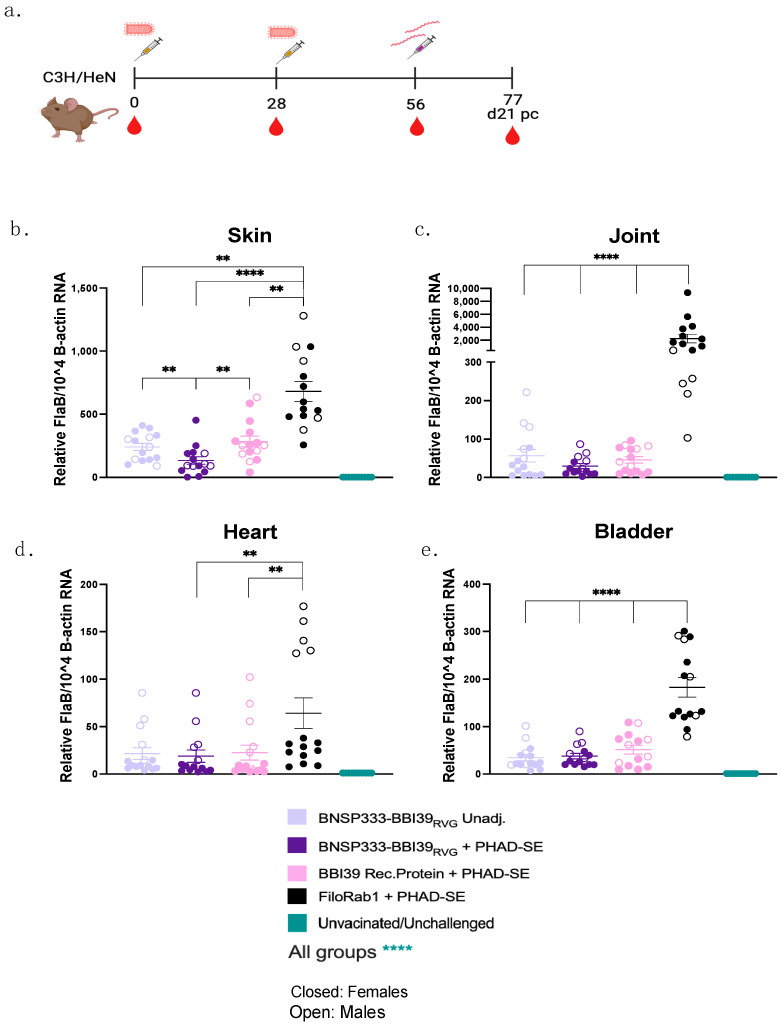
BNSP333-BBI39_RVG_ can successfully deplete *Borrelia burgdorferi* in mice. (**a**) Schedule of *Borrelia burgdorferi* challenge post-vaccination. Mice (5 female and 5 male) were challenged on day 56, and infected organs were collected on day 21 post challenge. Skin (**b**), joint (**c**), heart (**d**), and bladder (**e**) qPCRs after challenge. Statistics are represented by the color of the bars of each vaccinated group. FlaB from *Borrelia burgdorferi* is normalized to mouse beta-actin. Error bars represent the SEM. Statistics were calculated by one-way ANOVA with post hoc Tukey’s test of log-transformed data. *p* > 0.05 non-significant (ns), *p* < 0.0021 (**), *p* < 0.0001 (****).

**Figure 6 vaccines-12-00078-f006:**
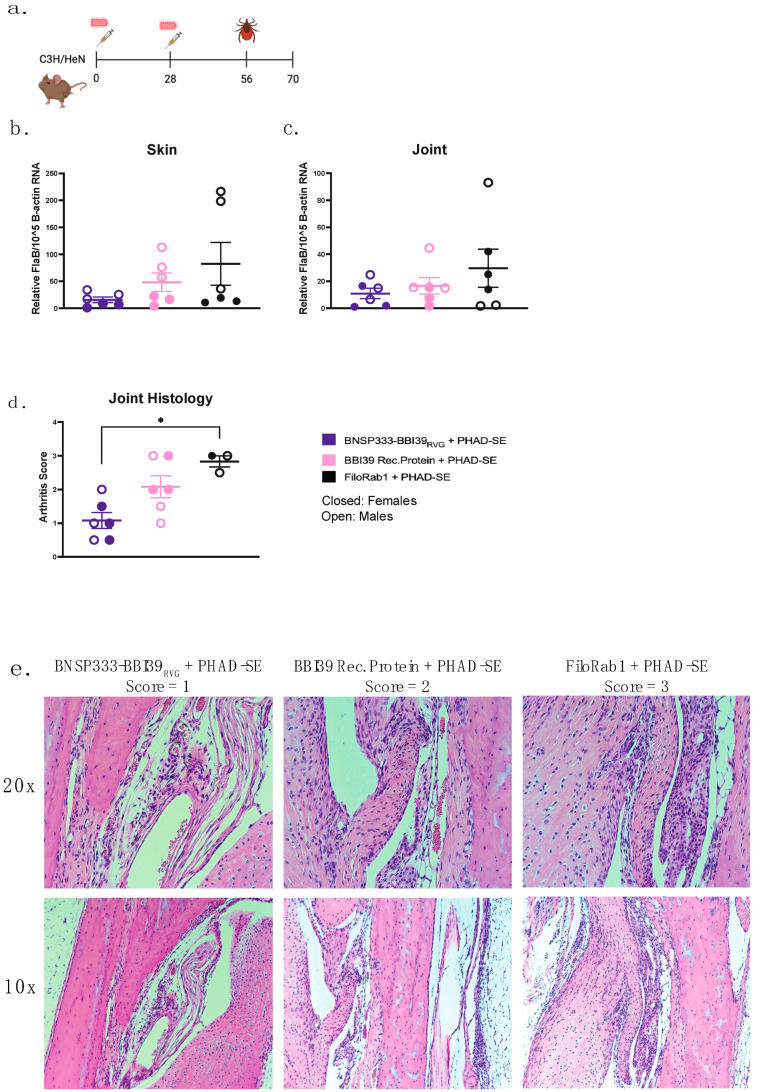
BNSP333-BBI39_RVG_ can deplete *Borrelia burgdorferi* in mice after an infected tick challenge. (**a**) Schedule of *Borrelia burgdorferi* challenge post-vaccination with infected ticks. Mice (3 female and 3 male) were challenged on day 56, and infected organs were collected on day 14 post challenge to perform skin (**b**) and joint (**c**) qPCR. FlaB from *Borrelia burgdorferi* is normalized to mouse beta-actin. Joints were subjected to histology analysis demonstrated by arthritis scores (**d**) and H&E staining (**e**). Scores were carried out in a blinded manner. Error bars represent the SEM. Statistics were calculated by one-way ANOVA with post hoc Tukey’s test of log-transformed data. *p* > 0.05 non-significant (ns), *p* < 0.0332 (*).

**Figure 7 vaccines-12-00078-f007:**
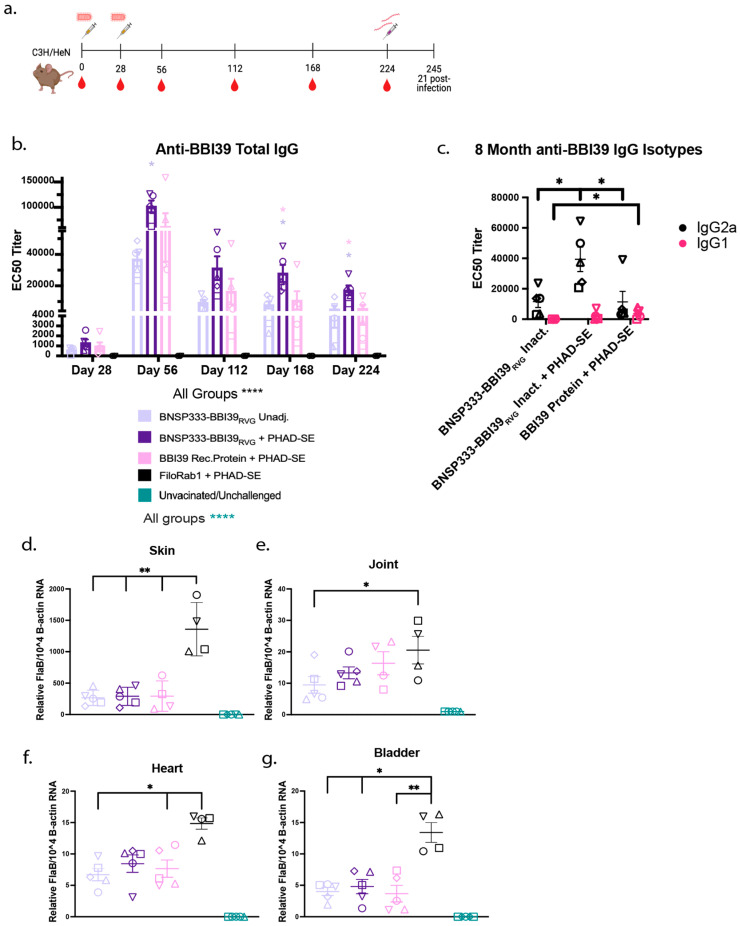
BNSP333-BBI39_RVG_ elicits anti-BBI39 antibodies and protects vaccinated mice long-term. (**a**) Vaccination and challenge schedule for long-term study. (**b**) anti-BBI39 total IgG antibody titers over time from day 28 to day 224 (8 months post initial vaccination) (*n* = 5 female mice/vaccine group). Various shapes represent different mice in that vaccinated group. (Statistics are represented by the color of the bars of each vaccinated group. (**c**) anti-BBI39 IgG isotypes 8 months post vaccination. qPCR of infected mouse organs: skin (**d**), joint (**e**), heart (**f**), and bladder (**g**). *FlaB* from *Borrelia burgdorferi* is normalized to mouse beta-actin. Error bars represent the SEM. Statistics were calculated by one-way ANOVA with post hoc Tukey’s test of log-transformed data. *p* > 0.05 non-significant (ns), *p* < 0.0332 (*), *p* < 0.0021 (**), *p* < 0.0001 (****).

## Data Availability

All data are available upon request from the corresponding authors.

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
