# Peer review of "The Development of a Rabies Virus-Vectored Vaccine against Borrelia burgdorferi, Targeting BBI39"

_vaccines, 2024, doi:10.3390/vaccines12010078_

Round 1

Reviewer 1 Report

Comments and Suggestions for Authors

The authors present a new vaccine consisting of a live-attenuated rabies virus recombinantly expressing the BBI39 antigen from Borrelia burgdorferi. The immunogenicity and challenge experiments are of good quality and help provide additional knowledge about rational vaccine design and generation of protective immune responses against this tick-borne illness. The manuscript is well-written and requires only minor revisions listed below: 

1.     Line 40: Please insert ‘of’ to make “during the early stage of LD”.

 2.     Line 93: The inclusion of “which is induced by borreliacidal antibodies” makes the sentence unclear. Please rephrase or remove. 

3.     Lines 317-328 are accidently italicized. 

4.     Please be sure citations are included for reagents produced by Pal lab to help demonstrate their utility and effectiveness.

5.     The X-axis legend for Figure 3E is difficult to read. 

6.     Figure 7 image quality is poor.

7.     Lines 641 and 642: Please change ‘wanning’ to ‘waning’.

8.     Reference 2 should be replaced with a more suitable citation. 

Author Response

The authors present a new vaccine consisting of a live-attenuated rabies virus recombinantly expressing the BBI39 antigen from Borrelia burgdorferi. The immunogenicity and challenge experiments are of good quality and help provide additional knowledge about rational vaccine design and generation of protective immune responses against this tick-borne illness. The manuscript is well-written and requires only minor revisions listed below:

  1. Line 40: Please insert ‘of’ to make “during the early stage of LD”.

This sentence was changed and this addition was not needed anymore.

  1. Line 93: The inclusion of “which is induced by borreliacidal antibodies” makes the sentence unclear. Please rephrase or remove.

We removed this part of the sentence.

  1. Lines 317-328 are accidently italicized.

Thank you, this has been corrected.

  1. Please be sure citations are included for reagents produced by Pal lab to help demonstrate their utility and effectiveness.

Yes, the citation (Singh et al, J Infect Dis. 215, 2017) is presented in the manuscript (reference 13).

  1. The X-axis legend for Figure 3E is difficult to read.

The X-axis font was made larger and updated in the text.

  1. Figure 7 image quality is poor.

Figure 7 was added again for better image quality.

  1. Lines 641 and 642: Please change ‘wanning’ to ‘waning’.

This was corrected and also changed in other areas of the paper.

  1. Reference 2 should be replaced with a more suitable citation.

The sentence was deleted which used reference 2.

Reviewer 2 Report

Comments and Suggestions for Authors

Dear Editor “Vaccine” Journal,

I have carefully reviewed the manuscript entitled "Development of a Rabies virus-vectored vaccine against Borrelia burgdorferi targeting BBI39" and appreciate the valuable research the authors have conducted. The research would be a great contribution to the development of a vaccine against Lyme borreliosis. I have raised some general comments. The paper is suitable for publication in the Vaccine Journal after addressing the following comments.

1.       Overall, the abstract would be improved by providing more specific details about the methodology, results, and implications of the study. Including quantitative data, statistical analysis, and a clearer statement of objectives would enhance the clarity and impact of the abstract. The abstract does not mention any limitations or potential future directions for the research. Including these aspects would provide a more comprehensive view of the study.

2.       Including information on the sample size and experimental design would provide important insights into the robustness and generalizability of your findings. Please consider including these details to enhance the transparency of your research.

3.       Please consider adding a clear statement of the study's objectives and hypothesis to provide a focused context for the results presented. This will help readers understand the research question being addressed and the specific goals of your study.

4.       To enhance the credibility of your results, I recommend including specific details and data on the efficacy of the vaccine in terms of antibody titers, infection rates, or disease severity in the challenged mice. This would provide a clearer understanding of the vaccine's effectiveness in neutralizing Borrelia and reducing Lyme disease pathology.

5.       It would be useful to briefly mention the potential implications or applications of your developed vaccine. Are there any plans for further testing or clinical trials? Addressing these points would give readers a sense of the broader significance and impact of your work.

I appreciate the scientific contribution your study makes to the development of a Lyme disease vaccine. Addressing these comments will strengthen your manuscript and improve its overall clarity and impact. I look forward to seeing the revised version of the manuscript.

Sincerely

Comments on the Quality of English Language

The only minor correction is needed

Author Response

  1. Overall, the abstract would be improved by providing more specific details about the methodology, results, and implications of the study. Including quantitative data, statistical analysis, and a clearer statement of objectives would enhance the clarity and impact of the abstract. The abstract does not mention any limitations or potential future directions for the research. Including these aspects would provide a more comprehensive view of the study.

Thank you for your suggestions. We added more information on the methogology, results, and conclusions from this study to the abstract (in red):

“BBI39 was utilized as a recombinant protein vaccine previously and was protective in challenge; therefore, we decided to utilize this protective antigen in a rabies virus-vectored vaccine against Borrelia burgdorferi. To incorporate BBI39 into the RABV virion, we generated a chimeric BBI39 antigen, BBI39RVG, by fusing it with the final amino acids of the RABV glycoprotein by molecular cloning and viral recovery with reverse transcription genetics. Here, we demonstrated that BBI39RVG antigen was incorporated into the RABV virion by immunofluorescence and western blot analysis. Mice vaccinated with our BPL inactivated RABV-BBI39RVG (BNSP333-BBI39RVG) vaccine induced high amounts of BBI39-specific antibodies, which were maintained long-term, up to eight months post-vaccination. The BBI39 antibodies neutralized Borrelia in vaccinated mice when challenged with Borrelia burgdorferi by either syringe injection or infected ticks and reduced the Lyme disease pathology arthritis in infected mouse joints. Overall, the RABV-based LD vaccine induced higher and long-term antibodies compared to the recombinant protein vaccine. This resulted in lower borrelial RNA in RABV-based vaccinated mice compared to recombinant protein vaccinated mice. The results of this study indicate the successful use of BBI39 as a vaccine antigen and RABV as a vaccine vector for LD.” (lines 16-30)

  1. Including information on the sample size and experimental design would provide important insights into the robustness and generalizability of your findings. Please consider including these details to enhance the transparency of your research.

We agree with your statement. We included sample size of 5 mice per vaccine group for live and inactivated immunization experiments (figure 3b-c) (line 422) and neutralization experiments (figure 4a-b) (line 505). However we utilized 10 mice per group (figure 3d-e) (line 454 and 477) and (figure 5) (line 516 and 541) when we wanted to include both 5 female and 5 males per group. For tick challenge we included sample size of 6 mice (figure 6) (line 552 and 567), 3 male and 3 female. We wanted to include repeat experiments with male and female mice to exclude sex as a scientific variable.

  1. Please consider adding a clear statement of the study's objectives and hypothesis to provide a focused context for the results presented. This will help readers understand the research question being addressed and the specific goals of your study.

Thank you for the suggestion. We added a couple of statements to clearly define the study’s objectives and hypothesis to the last paragraph of the introduction:

“In this study, we utilized the attenuated RABV vaccine vector with the borrelial outer surface protein BBI39. We believed that the addition of the RABV vaccine vector would induce high antibody titers that last long-term more than a recombinant protein vaccine. BBI39 is an ideal vaccine antigen because it has been shown to protect against Bb infection as a recombinant protein immunization. Therefore, when BBI39 is vectored with RABV, there will be better immunogenicity and protection against Bb. We demonstrate that the RABV vector successfully incorporated the chimeric BBI39RVG antigen into the RABV virion, which is necessary for antibody production. Mice immunized with our candidate vaccine, BNSP333-BBI39RVG, induced high and long-term anti-BBI39 antibody titers with a Th-1 biased immune response compared to the recombinant protein vaccine. In addition, we studied the adjuvant effects of PHAD-SE to determine if this adjuvant is ideal for an LD vaccine. Finally, we show that vaccinated mice inhibited Bb infection during both syringe inoculum and Bb-infected tick challenge. We show that the RABV vectored BBI39RVG vaccine is an ideal vaccine candidate against LD.” (Lines 95-108)

  1. To enhance the credibility of your results, I recommend including specific details and data on the efficacy of the vaccine in terms of antibody titers, infection rates, or disease severity in the challenged mice. This would provide a clearer understanding of the vaccine's effectiveness in neutralizing Borrelia and reducing Lyme disease pathology.

Thank you for your suggestion. We mentioned the increase of antibody titers between days (example: day 14 to 28, 28 to 56) but we did not compare antibody titers between groups. We added:

“Overall, inactivated BNSP333-BBI39RVG + PHAD-SE induced a 1.5-fold higher anti-BBI39 IgG responses than recombinant protein BBI39 + PHAD-SE immunized mice (Figure 3d).” (lines 452-453). We also describe fold differences in antibody titers in male and female mice between vaccine groups (lines 455-456). Finally, we added “BNSP333-BBI39RVG + PHAD-SE induced a 2-fold higher IgG2a response than the unadjuvanted and recombinant protein immunized groups (Figure 3e).” (line 465)

For the tick challenge we added the fold change in Bb burden in vaccinated mice (in red):

“The data in Figure 6b-c show that BNSP333-BBI39RVG immunization reduced Bb burden in mouse skin (5-fold) and joints (3-fold) compared to the control group, FiloRab1, during infected tick challenge. The RABV-based vaccine reduced Bb burden more than the recombinant protein vaccine, with almost 2-fold more in the skin and 1-fold in the joints. In addition, the mouse joints from tick challenge were analyzed for arthritis by histopathology. BNSP333-BBI39RVG vaccinated mouse joints showed significantly lower arthritis scores than FiloRab1 vaccinated mice (Figure 6d-e). Recombinant protein immunized mice did not significantly deplete arthritis in vaccinated mice. Therefore, we observed that BNSP333-BBI39RVG + PHAD-SE can successfully reduce LD pathology in infected tick challenge.” (lines 555-564).

  1. It would be useful to briefly mention the potential implications or applications of your developed vaccine. Are there any plans for further testing or clinical trials? Addressing these points would give readers a sense of the broader significance and impact of your work.

Thank you for the suggestion. We mention future experiments in the last paragraph of the discussion (lines 691-713). We mention further analyzing the vaccine in pre-clinical testing such as various mouse models (line 692), disease pathogenesis (lines 695-701) and challenge against different strains of Borrelia seen in Europe (line 704-706). Also, improving the vaccine with other antigens such as OspA, OspC, etc., since we did not observe sterile immunity in this study. (lines 708-714).

We also added a conclusion section at the end of the paper:

Conclusions

            In this study, we created a vaccine, BNSP333-BBI39-RVG, which targets BBI39, a surface protein on Bb. This study is the second to utilize BBI39 as a vaccine antigen and the first to utilize RABV as a vaccine vector for LD. We demonstrated that to produce anti-bodies against BBI39, the bacterial antigen must be incorporated into the RABV virion with the RVG tail to create the chimeric protein BBI39RVG. Both anti-BBI39 and an-ti-RABV-G IgG antibodies are elicited against this vaccine, thereby containing neutral-izing functions against both Bb and RABV in vitro. Thus, vaccinated mice can successfully deplete Bb during syringe and tick challenge. In addition, RABV vectored vaccines can induce greater immunity and protect against Bb long-term, up to 8 months post-immunization, in mice. Long-term protection is necessary when creating a successful vaccine.

We showed the advantages of a viral-vectored vaccine compared to recombinant protein vaccines. Viral-vectored vaccines induce greater type-1 immunity while recom-binant protein vaccines produce a more balanced type-1/2 immunity. A type-1 biased from the viral vector can kill Bb more efficiently than recombinant protein vaccinated mice. In addition, PHAD-SE is an ideal adjuvant in the formulation of an LD vaccine. We saw higher depletion of Bb in PHAD-SE vaccinated mice compared to unvaccinated mice. In vaccinated mice with adjuvant, we observed higher antibody production overall. The results from this study will help further research to develop an effective human LD vaccine. (lines 715-732)

We also mention our conclusions in the beginning of the discussion. (lines 612-623).

For the editing of the English language: We had an editor/scientific writer at Thomas Jefferson University read over the paper and correct any language errors.

Reviewer 3 Report

Comments and Suggestions for Authors

General comments:

1.    There is no obvious reason why BBI39 was selected as a vaccine antigen. The paper also lacks important details about BBI39, including the expression of BBI39 in the tick and during mammalian infection, its role on the Bb surface, the location of the gene, its degree of variability and if it is found in all Bb.

2.    The autoimmunity question about Osp A has been resolved and even Dr. Steere has repudiated his own hypothesis. Sequence homology does not predict T cell cross reactivity. The authors should be aware of this and not perpetuate false information. (Gross, D. M. et al. Identification of LFA-1 as a candidate autoantigen in treatment-resistant Lyme arthritis. Science 281, 703-706, doi:10.1126/science.281.5377.703 (1998).

Benoist, C. & Mathis, D. Autoimmunity provoked by infection: how good is the case for T cell epitope mimicry? Nat Immunol 2, 797-801, doi:10.1038/ni0901-797 (2001).

Mason, D. A very high level of crossreactivity is an essential feature of the T-cell receptor. Immunol Today 19, 395-404, doi:10.1016/s0167-5699(98)01299-7 (1998).

Hemmer, B. et al. Predictable TCR antigen recognition based on peptide scans leads to the identification of agonist ligands with no sequence homology. J Immunol 160, 3631-3636 (1998).)

3.    There are recent reviews of the history of LYMErix that the authors should also read.

4.    The clinical picture of Lyme disease is misrepresented and these misconceptions that need to be corrected. “Flu like illness” implies nasal and other respiratory signs and symptoms. Bb was not cause this. “Viral like illness” is more appropriate. Erythema migrans (EM) is quite variable and the “classic bulls eye” is actually not the most common presentation. (Schotthoefer AM, Green CB, Dempsey G, Horn EJ. The Spectrum of Erythema Migrans in Early Lyme Disease: Can We Improve Its Recognition? Cureus. 2022 Oct 25;14(10)). In addition, there are other cause of EM like lesions other than Bb. You need to correctly define EM.

Bb infection can affect multiple organ systems, but overt clinical manifestations are by no means inevitable. Also, symptoms are quite different than signs. Do not confuse the two. Over diagnosis is much more common than under diagnosis. Historical, in the United States, musculoskeletal manifestations have been emphasized as the major feature of late disease. Arthralgias and true arthritis are commonly conflated. This erroneously inflates the incidence of Lyme arthritis when in truth, the incidence of actual arthritis is low. In a recent North American study of 1230 LD patients, the overall incidence of arthritis was only .028%. In that study, of the 475 patients with Late LD only 35 (7.4%) manifested arthritis, while 440 (92.6%) only had arthralgias. (Johnson, K. O. et al. Clinical manifestations of reported Lyme disease cases in Ontario, Canada: 2005-2014. PLoS One 13, (2018).)

5.    Bb infection is treatable and curable at any stage. Continued symptoms or even signs do not necessarily indicate continued infection. (Carlson, D. et al. Lack of Borrelia burgdorferi DNA in synovial samples from patients with antibiotic treatment-resistant Lyme arthritis. Arthritis Rheum 42, 2705-2709, (1999)).

6.    The results seem to indicate that the quality of Bb was reduced in the vaccinated mice but not eliminated. If as some suspect, the the ongoing symptoms or signs associated with LD are immunologically mediated wouldn’t continued presences of even a small amount of Bb continue to be an isssue?

Specific comments:

1.    Line 28, The etiologic agent is Bb in North America.

2.    Line 33. EM is present in some studies as frequently as 90%. If the definition is a “bulls eye”, it will be much less.

3.    Line 38, Severe joint pain is rare. It is usually mild; heart palpitations, are common in the general population and not usually associated with Lyme carditis. The most common cardiac manifestation is heart block.

4.    Line 41.  PTLDS maybe chronic but it has not been proven to be a chronic inflammatory disease. Its mechanisms are poorly defined.

5.    Line 274. Does the cultured Bb express BBI39?

Author Response

General comments:

  1. There is no obvious reason why BBI39 was selected as a vaccine antigen. The paper also lacks important details about BBI39, including the expression of BBI39 in the tick and during mammalian infection, its role on the Bb surface, the location of the gene, its degree of variability and if it is found in all Bb.

In the paragraph starting at line 68, we mention how we wanted to develop a non-OspA LD vaccine, therefore we utilized BBI39 as our vaccine antigen. We describe how BBI39 is an outer surface protein which was previously immunogenic as a recombinant protein vaccine and depleted borrelial load/LD pathogenesis (lines 69-78). We also mention that BBI39 is produced on Bb while residing in the tick and early host infection (lines 74-75).

The role of BBI39 on the surface of Bb is currently unknown. This is mentioned in the previous study on BBI39: Singh P, et. al. Borrelia burgdorferi BBI39 Paralogs, Targets of Protective Immunity, Reduce Pathogen Persistence Either in Hosts or in the Vector. J Infect Dis. 2017 Mar 15;215(6):1000-1009. doi: 10.1093/infdis/jix036. PMID: 28453837; PMCID: PMC5407057.

We have now added it in line 70.

We added the description of the gene location: “BBI39 is an outer surface protein produced on Bb with unknown function [13]. BBI39 is in the paralogous gene family (pgf), found on plasmid lp54 in Bb. Pgfs are differentially expressed and regulated by temperature, pH, and various other intrinsic factors of the tick or mammalian hosts. BBI39 is expressed on the surface of Bb within the tick and early host infection in the skin.” (lines 70-73)

The Pal lab have shown that BBI39 is variably produced by all tested and diverse isolates of B. burgdorferi, such as isolate B31, N40 and 297 (Singh et al, J Infect Dis. 215, 2017).

  1. The autoimmunity question about Osp A has been resolved and even Dr. Steere has repudiated his own hypothesis. Sequence homology does not predict T cell cross reactivity. The authors should be aware of this and not perpetuate false information. (Gross, D. M. et al. Identification of LFA-1 as a candidate autoantigen in treatment-resistant Lyme arthritis. Science 281, 703-706, doi:10.1126/science.281.5377.703 (1998).

Benoist, C. & Mathis, D. Autoimmunity provoked by infection: how good is the case for T cell epitope mimicry? Nat Immunol 2, 797-801, doi:10.1038/ni0901-797 (2001).

Mason, D. A very high level of crossreactivity is an essential feature of the T-cell receptor. Immunol Today 19, 395-404, doi:10.1016/s0167-5699(98)01299-7 (1998).

Hemmer, B. et al. Predictable TCR antigen recognition based on peptide scans leads to the identification of agonist ligands with no sequence homology. J Immunol 160, 3631-3636 (1998).)

Thank you for the insightful information. We corrected how we described the OspA vaccines and included your reference in the paper (in red):

“However, there was a reported linkage of arthritis development in vaccinated patients with treatment resistant Lyme arthritis [11]. (line 62-63)

“While OspA is protective, it was found to contain a similar epitope to the human leukocyte function-associated antigen-1 (hLFA1), indicating the possibility of arthritis development in patients with treatment-resistant Lyme arthritis [11].” (lines 625-627).

  1. There are recent reviews of the history of LYMErix that the authors should also read.

We have now included a recent review as a reference in this paper (reference 40): Wormser GP. A brief history of OspA vaccines including their impact on diagnostic testing for Lyme disease. Diagn Microbiol Infect Dis. 2022 Jan;102(1):115572.

  1. The clinical picture of Lyme disease is misrepresented and these misconceptions that need to be corrected. “Flu like illness” implies nasal and other respiratory signs and symptoms. Bb was not cause this. “Viral like illness” is more appropriate. Erythema migrans (EM) is quite variable and the “classic bulls eye” is actually not the most common presentation. (Schotthoefer AM, Green CB, Dempsey G, Horn EJ. The Spectrum of Erythema Migrans in Early Lyme Disease: Can We Improve Its Recognition? Cureus. 2022 Oct 25;14(10)). In addition, there are other cause of EM like lesions other than Bb. You need to correctly define EM.

Bb infection can affect multiple organ systems, but overt clinical manifestations are by no means inevitable. Also, symptoms are quite different than signs. Do not confuse the two. Over diagnosis is much more common than under diagnosis. Historical, in the United States, musculoskeletal manifestations have been emphasized as the major feature of late disease. Arthralgias and true arthritis are commonly conflated. This erroneously inflates the incidence of Lyme arthritis when in truth, the incidence of actual arthritis is low. In a recent North American study of 1230 LD patients, the overall incidence of arthritis was only .028%. In that study, of the 475 patients with Late LD only 35 (7.4%) manifested arthritis, while 440 (92.6%) only had arthralgias. (Johnson, K. O. et al. Clinical manifestations of reported Lyme disease cases in Ontario, Canada: 2005-2014. PLoS One 13, (2018).)

Thank you for this information and suggested references. We corrected the clinical picture of lyme disease and added the reference suggested:

“An erythema migrans rash, a clinical skin lesion, can appear on the skin due to the spread of Bb from the initial bite site [2]. Symptoms of early-stage LD mimic a viral-like illness and include fever, headache, fatigue, muscle and joint aches, and swollen lymph nodes [3]. However, 20-30% of patients do not display the rash, the rash can vary in morphological features, and patients can test negative for Bb with a rash present, which challenges clinicians in making an LD diagnosis [2,4].” (lines 38-44).

  1. Bb infection is treatable and curable at any stage. Continued symptoms or even signs do not necessarily indicate continued infection. (Carlson, D. et al. Lack of Borrelia burgdorferi DNA in synovial samples from patients with antibiotic treatment-resistant Lyme arthritis. Arthritis Rheum 42, 2705-2709, (1999)).

Thank you, we included this in our background information about the antibiotics and PTLDS. “Currently, the only treatment for LD is antibiotics; however, after treatment and Bb clearance, patients can still develop Post-treatment Lyme Disease Syndrome (PTLDS), a chronic disease in patients previously diagnosed with LD [6]. These complications indicate a need for a human LD vaccine.” (line 47-51)

  1. The results seem to indicate that the quality of Bb was reduced in the vaccinated mice but not eliminated. If as some suspect, the the ongoing symptoms or signs associated with LD are immunologically mediated wouldn’t continued presences of even a small amount of Bb continue to be an isssue?

Yes this you are correct. This is one of the reasons why we looked into the disease pathogenesis with histology. We see the disease pathogenesis of LD was depleted in vaccinated mice with prescence of Bb (Figure 6). However, we do mention the small amounts of Bb still present in the vaccinated is a pitfall of this vaccine. We mention “Further preclinical testing to determine the inhibition of disease pathogenesis, such as arthritis and carditis, should be completed. This includes long-term challenge experiments or keeping vaccinated and challenged mice for longer periods than 21 days post-challenge. This could show if the lower borrelial burdens in vaccinated mice could continue to prevent LD pathogenesis.” (Lines 695-699). We recognize small amount of Bb present could continue to be an issue and would be studied in future experiments with this vaccine. We also mention “Since LD is highly inflammatory, the immune profile of vaccinated and unvaccinated mice, such as cytokine profiles after vaccination and challenge, should also be studied.” (Lines 699-701).

Specific comments:

  1. Line 28, The etiologic agent is Bb in North America.

Corrected: “The etiologic agent of LD in North America is the spirochete pathogen Borrelia burgdorferi (Bb)” Line 36-37

  1. Line 33. EM is present in some studies as frequently as 90%. If the definition is a “bulls eye”, it will be much less.

Corrected by taking out the “Bull’s eye” appearance: “An erythema migrans rash, a clinical skin lesion, can appear on the skin due to the spread of Bb from the initial bite site [2].” Lines 38-40

  1. Line 38, Severe joint pain is rare. It is usually mild; heart palpitations, are common in the general population and not usually associated with Lyme carditis. The most common cardiac manifestation is heart block.

Corrected: “This can cause complications including arthritis with joint pain and swelling, heart block, and inflammation of the brain and spinal cord [4].” Line 47

  1. Line 41. PTLDS maybe chronic but it has not been proven to be a chronic inflammatory disease. Its mechanisms are poorly defined.

We removed the “inflammatory” description and described PTLDS as “a chronic disease in patients previously diagnosed with LD [6].” Lines 49-50

  1. Line 274. Does the cultured Bb express BBI39?

Yes, cultured Bb expresses BBI39. Needle injections with cultured Bb and in vitro borreliacidal assays have been done with anti-BBI39 vaccinated mouse sera by the Pal lab. Also, BBI39 is produced in the tick and early host infections, therefore the culture temperature is optimal for BBI39 production.

Round 2

Reviewer 3 Report

Comments and Suggestions for Authors

The authors only partly addressed my comments. 

They failed to meaningfully address the question of hLFA1 cross reactivity with OspA. I supplied references if read would lead to a better understand of T cell epitopes, their binding to the T cell receptor and T cell epitope cross reactivity. It is very clear that sequence homology is not predictive of T cell epitope cross reactivity and that to state otherwise is bad science. The Gross paper used sequence homology to predict T cell cross reactivity. Therefore, its conclusions are totally invalid. It is wrong to perpetuate this fallacy. As I previously pointed out, even its senior author, Allen Steer has repudiated that study. Also, sticking in the Wormser paper misses the point. Unlike the reference used, Dattwyler, R.J., Gomes-Solecki, M. The year that shaped the outcome of the OspA vaccine for human Lyme disease. npj Vaccines 7, 10 (2022) addresses the issue of cross reactivity and is more on point. 

An issue that is not adequately addressed is that the proposed vaccine is that the vaccine only reduced borrelia load and did not prevent infection. 

Author Response

They failed to meaningfully address the question of hLFA1 cross reactivity with OspA. I supplied references if read would lead to a better understand of T cell epitopes, their binding to the T cell receptor and T cell epitope cross reactivity. It is very clear that sequence homology is not predictive of T cell epitope cross reactivity and that to state otherwise is bad science. The Gross paper used sequence homology to predict T cell cross reactivity. Therefore, its conclusions are totally invalid. It is wrong to perpetuate this fallacy. As I previously pointed out, even its senior author, Allen Steer has repudiated that study. Also, sticking in the Wormser paper misses the point. Unlike the reference used, Dattwyler, R.J., Gomes-Solecki, M. The year that shaped the outcome of the OspA vaccine for human Lyme disease. npj Vaccines 7, 10 (2022) addresses the issue of cross reactivity and is more on point. 

Thank you for the insightful review. This cleared a lot of things up as to why you ask us to change the description of OspA. We added these changes to the introduction and discussion in blue (red was first round responses):

“Previously, the FDA approved an LD vaccine, LYMErix, a recombinant OspA protein vaccine adjuvanted with alum [10]. OspA has been shown to protect against Bb and other strains of Borrelia. LYMErix decreased LD rates by 68% within the first year and 92% efficacy in the second year on the market [11]. A couple years after the vaccine was on the market, there was a reported linkage of arthritis development in vaccinated patients [12]. Later, this linkage was refuted, however, data demonstrated that patients with treatment-resistant Lyme arthritis contain the gene for MHC class II HLA DR4B1*0401 or DR4B1*0101 which reacts to the human leukocyte function-associated antigen-1 (hLFA-1), producing an inflammatory response in the joints. OspA contains a similar epitope to hLFA-1(OspA165-183) [12]. Therefore, when these patients are infected with Bb and/or vaccinated with OspA antigen, their T cells which already react to hLFA-1 cross-react with the OspA165-183 peptide, producing IFNl and an inflammatory response in the joint [11]. In addition, other factors including low efficacy, need for boosters for sufficient neutralizing antibody titers, and lack of studies in children were reasons why the vaccine was taken off the market[11]. OspA remains a target antigen in LD vaccines such as in Vanguard crLyme [13], a dog vaccine, and VLA15, a human vaccine developed by Pfizer currently in phase III clinical trials (NCT05477524) [7].” (Lines 58-74)

“In previous studies, OspA was the most utilized vaccine antigen [7, 13, 35, 39, 40]. While OspA is protective, it was found to contain a similar epitope to the human leukocyte function-associated antigen-1 1 (hLFA-1aL322-340) (OspA165-183) [12]. It was discovered that patients with treatment-resistant arthritis contain the MHC class II HLA DR4B1*0401 or DR4B1*0101 gene. Therefore, their T cells cross-react to the human leukocyte function-associated antigen-1 (hLFA-1aL322-340) and OspA165-183, producing an inflammatory response in the joints. In addition, other factors including low efficacy, need for boosters for sufficient neutralizing antibody titers, and lack of studies in children were reasons why the vaccine was taken off the market [11]. Low efficacy and need for boosters could correlate with the recombinant protein vaccine platform used in LymeRix, also seen with the recombinant protein immunization from the long-term experiments in this study and a previous study [41]. Ultimately, the company removed the vaccine from the market in 2002 [42]. To prevent these complications from arising, we utilized BBI39, a different surface protein on Borrelia previously seen to be protective against LD [14].” (Lines 633-644)

An issue that is not adequately addressed is that the proposed vaccine is that the vaccine only reduced borrelia load and did not prevent infection.  

We mention in our results that Borrelial load is significantly decreased in both syringe challenge and tick challenge (lines 530-532). We did add the statement “however still infected” in lines 539 and mentioned “reduced Bb load” in lines 565 and “Bb loads trended lower in all organs” line 599. We made sure to mention the Bb loads were reduced however the mice are still infected. Although these mice are infected, they also reduce LD pathology in BNSP333-BBI39RVG vaccinated mice (Figure 6). We also added “Thus, vaccinated mice can successfully deplete Bb load during syringe and tick challenge. Although these mice are still infected, the LD pathology was depleted.” Lines 735-737.

We also mention in the discussion “Finally, since we did not see sterile immunity with the BNSP333-BBI39RVG vaccine in either challenge experiment, future research could study different borrelial antigens in combination with the BBI39 vaccine.” Lines 720-722. We recognize this vaccine reduces Bb load and does not prevent infection.

Round 3

Reviewer 3 Report

Comments and Suggestions for Authors

The revision, "Later, this linkage was refuted, however, data demonstrated that patients with treatment- resistant Lyme arthritis contain the gene for MHC class II HLA DR4B1*0401 or DR4B1*0101 which reacts to the human leukocyte function-associated antigen-1 (hLFA-1),  producing an inflammatory response in the joints. OspA contains a similar epitope to 66 hLFA-1(OspA165-183) [12]. Therefore, when these patients are infected with Bb and/or vaccinated with OspA antigen, their T cells which already react to hLFA-1 cross-react with 68 the OspA165-183 peptide, producing IFNland an inflammatory response in the joint [11]. In 6addition, other factors including low efficacy, need for boosters for sufficient neutralizing 70 antibody titers, and lack of studies in children were reasons why the vaccine was taken off the market [11]. " ,needs additional modifications.

1.     You are over stating things. There was a higher rate of MHC class II HLA DR4B1*0401 or 64 DR4B1*0101. They are not universal in the antibiotic refractory Lyme arthritis, and these MHC types are also associated with increased risk for a number of other inflammatory illnesses. Also, attributing joint inflammatory to hLFA-1 cross reactivity was only a hypothesis. It has not been proven. You should read the papers I suggested:

Benoist, C. & Mathis, D. Autoimmunity provoked by infection: how good is the case for T cell epitope mimicry? Nat Immunol 2, 797-801, doi:10.1038/ni0901-797 (2001).

Mason, D. A very high level of crossreactivity is an essential feature of the T-cell receptor. Immunol Today 19, 395-404, doi:10.1016/s0167-5699(98)01299-7 (1998).

Hemmer, B. et al. Predictable TCR antigen recognition based on peptide scans leads to the identification of agonist ligands with no sequence homology. J Immunol 160, 3631-3636 (1998).

2.     MHC may bind to a peptide but it does not “react” to it.

Author Response

  1. You are over stating things. There was a higher rate of MHC class II HLA DR4B1*0401 or 64 DR4B1*0101. They are not universal in the antibiotic refractory Lyme arthritis, and these MHC types are also associated with increased risk for a number of other inflammatory illnesses. Also, attributing joint inflammatory to hLFA-1 cross reactivity was only a hypothesis. It has not been proven. You should read the papers I suggested:

We have simplified the statement in the Introduction about LymeRix. We do not want to focus too much on OspA in the introduction since this paper is about a BBI39 based vaccine. The point here is we do not have a LD vaccine on the market. However, this should remain a discussion point when talking about LD vaccine development. Green represents the changes in the 3rd round of revisions.

“Previously, the FDA approved an LD vaccine, LYMErix, a recombinant OspA protein vaccine adjuvanted with alum [10]. OspA has been shown to protect against Bb and other strains of Borrelia. LYMErix decreased LD rates by 68% within the first year and 92% efficacy in the second year on the market [11]. A couple years after the vaccine was on the market, there was a reported linkage of arthritis development in OspA vaccinated patients with treatment-resistant Lyme arthritis. In addition, vaccine sales declines and the vaccine was taken off the market [11]. OspA remains a target antigen in LD vaccines such as in Vanguard crLyme [12], a dog vaccine, and VLA15, a human vaccine developed by Pfizer currently in phase III clinical trials (NCT05477524) [7]. However, there is currently no LD vaccine on the market.” Lines 58-67